



# Can atmospheric chemistry deposition schemes reliably simulate stomatal ozone flux across global land covers and climates?

Tamara Emmerichs[1], Abdulla Al Mamun[2], Lisa Emberson[3], Huiting Mao[4], Leiming Zhang[2], Limei Ran[5], Clara Betancourt[6], Anthony Wong[7], Gerbrand Koren[8], Giacomo Gerosa[9], Min Huang[10], Pierluigi Guaita[9]

[1]Institute of Energy and Climate Systems, Troposphere (ICE-3), Forschungszentrum Jülich, Germany, now at Max-Planck-Institute for Meteorology, Germany

[2]Air Quality Research Division, Science and Technology Branch, Environment and Climate Change Canada, Toronto, Ontario M3H 5T4, Canada

[3]Environment & Geography Department, University of York, York, UK.

[4]Department of Chemistry, State University of New York College of Environmental Science and Forestry, Syracuse, NY, 13210, USA

[5]Resource Assessment Branch, USDA-NRCS-SSRA-RIAD

[6]AXA Konzern AG, Cologne, Germany

[7]Centre for Global Change Science, Massachusetts Institute of Technology, Cambridge, MA, United States of America

[8]Copernicus Institute of Sustainable Development, Utrecht University, Utrecht, the Netherlands

[9]Dep.t of Mathematics and Physics, Catholic University of the Sacred Heart, Brescia, Italy

[10]Earth System Science Interdisciplinary Center, University of Maryland, College Park, MD, USA

*Correspondence to*: Tamara Emmerichs (tamara.emmerichs@gmx.de)





**Abstract**
Over the past few decades, ozone risk assessments for vegetation have been developed based on stomatal $O_3$ flux
since this metric is more biologically meaningful than the traditional concentration-based approaches. However,
uncertainty remains in the ability to simulate stomatal $O_3$ fluxes accurately. Here, we investigate stomatal $O_3$
fluxes simulated by six common air pollution deposition models across various land cover types worldwide. The
Tropospheric Ozone Assessment Report (TOAR) database, a large collection of measurements worldwide,
provides hourly $O_3$ concentration and meteorological data which are used to drive the models at 9 sites. The
models estimated summertime $O_3$ deposition velocities of between 0.5 - 0.8 cm $s^{-1}$, mostly in agreement with the
literature. Simulations of canopy conductance ($G_{st}$) showed differences between models that varied by land
cover type with correlation coefficients of 0.75, 0.80 and 0.85 for forests, crops and grasslands. The model
differences were determined by especially soil moisture and VPD depending upon the model constructs. Finally,
the range of $POD_y$ simulations at each site across models was most in agreement for crops (3 to 11 mmol $O_3$ $m^{-2}$)
< forests (10 to 23 mmol $O_3$ $m^{-2}$) < grasslands (24 to 26 mmol $O_3$ $m^{-2}$). Nevertheless, ensemble model median
response estimates gave results consistent with the literature in terms of those sites where $O_3$ damage is most
likely to occur. Overall, this study is an important first step in developing and evaluating tools for broad-scale
assessment of $O_3$ impact on vegetation within the framework of TOAR phase II.
**1.  Introduction**
Elevated surface $O_3$ levels significantly damage vegetation due to the stomatal uptake of $O_3$ by canopy leaves.
Stomatal uptake of $O_3$ leads to plant tissue injury which in turn causes changes in metabolic functioning,
reducing photosynthesis and consequently plant growth and productivity (Mills et al., 2011; Emberson, 2020;
Ainsworth et al. 2012; Fuhrer et al., 2016; Grulke and Heath, 2020). Such damage can have significant impacts
on crop yields and quality, leading to economic losses and impacting food security in regions already facing
scarcity (Avnery et al., 2011; Ainsworth et al. 2017; Ramya et al., 2023). There is an ever-growing body of
observational evidence demonstrating a variety of $O_3$ impacts on different ecosystems (crops, forests, grasslands)
in North America, Europe and more recently, Asia (Emberson 2020). Various indices assessing $O_3$ exposure to



vegetation have been developed over recent decades with the stomatal $O_3$ flux ($POD_y$; phytotoxic ozone dose
over a threshold y) index found to provide better estimates of $O_3$ risk to vegetation than the more commonly
used concentration-based exposure approaches (e.g., Accumulated Ozone over Threshold (AOT); growing
season daylight mean $O_3$ concentration (M7, M12) (Mills et al., 2011; Avnery et al., 2011). A global overview of
spatial distribution and trends using concentration-based metrics was provided in the first Tropospheric Ozone
Assessment Report (TOAR) by Mills et al. (2018). During TOAR phase II (TOAR-II), here we conduct a flux-
based analysis to ensure the most up-to-date vegetation metrics are provided through this community effort.
$O_3$ dry deposition to vegetation is in part determined by canopy-level $O_3$ concentrations. A significant fraction of
$O_3$ uptake occurs through the plant stomata with the remainder depositing on plant cuticular surfaces and the
under-storey vegetation and soil. The stomatal contribution can vary between 50 and 80%, depending on the
factors controlling the partitioning of stomatal and non-stomatal uptake (e.g., Huang et al., 2022; Wong et al.,
2022; Clifton et al., 2023). As such, quantifying canopy stomatal conductance is important for assessing the
mass balance of atmospheric $O_3$ concentrations and its potential damage to vegetation. Both stomatal and non-
stomatal processes can vary with environmental conditions such as humidity, solar radiation, temperature and
$CO_2$ concentration as well as vegetation type and density (Clifton et al., 2020a). The occurrence of soil water
deficit can also play a crucial role where soil water stress induces stomatal closure (Lin et al., 2020; Huang et al.,
2022). There are two commonly used stomatal conductance ($g_s$) models - the empirical, multiplicative approach
first developed by Jarvis (1976) and the semi-mechanistic coupled net photosynthesis-stomatal conductance
models (Anet-gs). The common Jarvis-type models (e.g. Emberson et al., 2000; Ganzeveld et al., 1995; Zhang et
al. 2003), widely applied due to their simplicity and computational efficiency, correct a prescribed maximum
stomatal conductance with the multiplication of different environmental factors (e.g., temperature, light, soil
water and atmospheric moisture). The $A_{net}$-$g_s$ models couple $g_s$ to plant photosynthesis by calculating the net
assimilation of $CO_2$ and estimating gs based on the resulting supply and demand of $CO_2$ (Farquhar et al., 1980;
Goudriaan et al., 1985; Ball et al., 1987). $A_{net}$-$g_s$ models involve multiple non-linear dependencies on soil water,
humidity and temperature, among other factors defined by measurement constraints (Ball 1987; Leuning et al.,
1997). Heterogeneity of stomatal deposition estimates over different land cover types is anticipated, but model
uncertainty depends on the representation of the deposition mechanisms, model parameterisation and
meteorological inputs (Hardacre et al., 2015; Clifton et al., 2020b; Huang et al., 2022; Khan et al., 2024).
In this study, the stand-alone version of six $O_3$ deposition schemes, commonly used in climate or air quality
models, are assessed with a focus on their stomatal uptake portion and resulting $POD_y$ calculation. Using



concurrent $O_3$ concentration and meteorological variable measurement data from the TOAR database enables us
to conduct a detailed intercomparison of multiple deposition schemes by avoiding uncertainties arising from
using different input data. For this study, various sites have been selected to represent different land cover types
and climate regimes around the globe, focusing on sites where observational data are available for $O_3$
concentration. By assessing the model estimates of stomatal $O_3$ deposition at these different sites, we aim to
identify key differences in model formulation and parameterisation that influence estimates of stomatal $O_3$ flux
and consequent $POD_y$. The estimation of the stomatal uptake from water flux measurements taken from the
FLUXNET database provides an additional observational constraint as well as an uncertainty estimate at each
site.
Furthermore, sensitivity simulations allow us to investigate the variability of stomatal $O_3$ deposition and plant
damage with key input parameters and land cover characteristics. Post hoc, plant damage will be calculated
offline based on the $POD_y$ simulated by different models and flux-response relationships, where appropriate.
Ultimately, we aim to understand the key factors driving stomatal $O_3$ flux and thus $POD_y$ and assess the $O_3$-
induced potential for vegetation damage for different land cover types and global regions.

**2. Methodology**
**2.1 Meteorological and $O_3$ data from the TOAR-II database**
The TOAR-II database (from now on TOAR) contains harmonised measurements of surface $O_3$ and its important
precursors and key meteorological variables that can impact $O_3$ concentrations and stomatal $O_3$ uptake. As one
of the largest collections of quality-controlled air pollution measurements in the world, it comprises ground-
based station measurements of $O_3$ concentration at more than 22905 sites globally which cover different periods
between 1974 and 2023. These have been collected from different $O_3$ monitoring networks (e.g., Clean Air
Status and Trends Network, CASTNET), harmonised and synthesised to enable uniform processing. The data
were selected for inclusion in the TOAR database based on an extended quality control; e.g., sites where the
measurement technique changed with time have been excluded. Data errors remain but have been shown to have
a minor impact (Schultz et al., 2017). The total uncertainty in modern $O_3$ measurements is estimated to be $< 2$
nmol/mol (Tarasick et al., 2018). The meteorological data (irradiance, air temperature, relative humidity,



precipitation, air pressure, and wind speed) in the database stems from the fifth generation of ECMWF
reanalysis (ERA5) for global climate (Hersbach et al., 2020). Data re-initialisation (of precipitation and
radiation, Copernicus Climate Change Service, 2017) is bridged by (linear) interpolation. The Leaf Area Index
(LAI) data in the database stems from the MODerate resolution Imaging Spectroradiometer (MODIS). TOAR
data is freely, and openly available through a graphical user interface and a representational State Transfer
interface (https://toar-data.fz-juelich.de/api/v2/, last access: 01.11.2024). The TOAR data centre team is
committed to the Findability, Accessibility, Interoperability, and Reusability principles (Wilkinson et al., 2016).
The centre aims to achieve the highest standards regarding data curation, archival, and re-use (Schröder et al.,
2021). In this study, additional meteorological ERA5 data required by some models were extracted from the
MeteoCloud server (https://datapub.fz-juelich.de/slcs/meteocloud/index.html) at Forschungszentrum Jülich.

## 2.2 Observation-constrained stomatal conductance

To compare the modelled stomatal conductance with observational information, we prepared model input data at
two sites (Hyytiälä, Harvard Forest) from the FLUXNET 2015 dataset (Pastorello et al., 2020), which is openly
available under the CC-BY-4.0 data usage licence. Additional vegetation information for the model input (i.e.,
LAI, canopy height, and crop calendar data) was provided by the site project investigators. Then, we used the
canopy-scale stomatal conductance dataset, SynFlux version 2 to estimate $G_{st}$ for two forest sites, US-Ha1 and
FI-Hyy. While in SynFlux version 1, canopy transpiration is assumed to be equal to total latent heat flux
SynFlux version 2 improved its previous estimations (Ducker et al., 2018) by using a machine-learning-based
method (Nelson et al., 2018) to partition total evapotranspiration into surface evaporation and canopy
transpiration. To train quantile random forest models to relate meteorological conditions with water use
efficiency (derived from water and carbon fluxes), periods with minimal surface wetness were chosen during the
growing season. These models were then used to back-calculate transpiration for the whole growing season.
Instead of the total latent heat flux, the resulting transpiration estimate was used as an input to the inverse
Penman-Monteith Equation, reducing the potential high bias in the stomatal conductance estimates in SynFlux
version 1.



**2.3 Summary of sites selected for deposition modelling**
Nine sites (Table 1) were selected for this modelling work accounting for the following factors: i. geographical
spread,including major continents with terrestrial vegetation; ii. land cover/use types, including the plant
functional types (PFTs) which are important in terms of economy, food security, or biodiversity and for which
we have fairly good knowledge of $O_3$ impacts ; iii. availability of meteorological and $O_3$ data from the TOAR
database; iv. availability of observational data describing stomatal conductance of water vapour ($g_{wv}$) estimated
from the FLUXNET measurements (Section 2.2); and v. location proximity to previous experiments that have
investigated $O_3$ impacts on vegetation that can help interpret our model results.
**Table 1. Sites selected for stomatal deposition modelling using data from the TOAR database grouped by**
**continent. Sites that also have FLUXNET data are denoted by 'FN' and those with SynFlux data are**
**denoted by 'SF'.**

| Site (TOAR station id, nearest FLUXNET site id) | Location, station altitude from TOAR | Köppen-Geiger climate classification | Vegetation details (LAI, canopy height in m) | Record (measurement heights in m) | References |
|---|---|---|---|---|---|
| Europe | | | | | |
| Hyytiälä, Finland (FI00621, FI-Hyy) FN & SF | 61.8611 °N, 24.2833 °E, 104 m | Dfc | LAI: 2.9 Height: 23.3 | $O_3$: 2014 (4) FLUXNET: 1996/04-2013/09 (14) | Chen et al. (2018); Junninen et al. (2009); Visser et al. (2021) |
| Grignon, France (FR04038, FR-Gri) FN | 48.5819 °N, 1.833 °E, 165 m | Cfb | LAI: 4.3 Height: 3.5 | $O_3$: 2013/2014 (3) FLUXNET: 2004-2014 (2) | Stella et al. (2013) |
| Castelporzian | 41.8894 °N, | Csa | LAI: 6.9 | $O_3$: | Gerosa et al. (2005, |





| o, Italy (IT0952A, IT-Cpz) | 12.266 °E, 19 m | | Height: 14.0 | 2013/2014 (19.7) FLUXNET: 2013/2014 (10) | 2009); Fares et al. (2009, 2012); personal communications with Silvano Fares |
|---|---|---|---|---|---|
| Asia | | | | | |
| Amberd, Armenia (AM0001R) | 40.3844 °N, 44.2605 °E, 2080 m | BSk (or Dfa) | LAI: 3.9 Height: 1.0 | $O_3$: 2009/2010 (3) | |
| Pha Din, Vietnam (VN0001R) | 21.5731 °N, 103.5157 °E, 1466.0 m | Cwa | LAI: 6.9 Height: > 10.0 | $O_3$: 2015-2017 (12) | Pieber et al. (2023); Bukowiecki et al. (2018); Yen et al. (2013) |
| North America | | | | | |
| Quabbin Reservoir/Harvard Forest tower, USA (25-015-4002, US-Ha1) FN & SF | 42.2985 °N, -72.3341 °E, 312 m | Dfb | LAI: 3.0 Height: 24.0 | $O_3$: 2010-2012 (2) FLUXNET: 1993-2012 (24) | Clifton et al. (2019, 2020b); Ducker et al. (2018) |
| Nebraska, USA (31-055-0032, US-Ne3) | 41.3602 °N, -96.0250 °E, 400 m | Dfa | LAI: 1.7 Height: 2.5 | $O_3$: 2010 (2) FLUXNET: 2013/04- (0.5) | Amos et al. (2005); Leung et al (2020) |
| South America | | | | | |




| | -12.0402 °N, -75.3209 °E, 3314 m | Cwb | LAI: 3.6 Height: 1.0 | $O_3$: 2015 (6) | |
| Huancayo, Peru (PE0001R) | | | | | |
| Africa | | | | | |
| Mt. Kenya, Kenya (KE0001G) | -0.062 °N, 37.297 °E, 3678.0 m | Aw | LAI: 4.2 Height: 1.0 | $O_3$: 2015 (unknown) | Henne et al. (2008a,b) |


**Table 2. Land cover type, species and growing season (where SGS = start of growing season and EGS =**
**end of growing season) by site. The equivalent land cover type and soil texture data used by the models**
**used in this study (Section 2.3) are also shown. MESSy does not consider different land cover types.**
**Models that do not consider soil type (i.e. do not include an estimate of soil moisture influence on stomatal**
**deposition) are marked with \*.**

| Station site: land cover type (species) and growing season | Web-D $O_3$ SE | TEMIR* | NOAH-GEM | ZHANG* | CMAQ |
|---|---|---|---|---|---|
| Hyytiälä, Finland: evergreen needleleaf forest (Scots pine) SGS=1, EGS=366 | evergreen needleleaf forest, loam | evergreen needleleaf boreal forest | evergreen needleleaf forest, organic material | evergreen needleleaf forest | evergreen needleleaf forest, silty loam (peat) |
| Grignon, France: crops (rapeseed and wheat) SGS=304, EGS=571 | winter wheat, loam | C3 crop | crops/grassland mosaic, silt loam | crops | crops (wheat), silty loam |





| | | | | | |
|---|---|---|---|---|---|
| Castelporziano, Italy: evergreen broadleaf forest (laurel, abutus, broad-leaved phillyrea, holm oak, pine) SGS=1, EGS=366 | evergreen broadleaf forest, loam | Evergreen broadleaf temperate forest | evergreen broadleaf forest, sandy loam | evergreen broadleaf forest | evergreen broadleaf forest, loamy sand |
| Amberd, Armenia: Grassland, mixed SGS=1, EGS=366 | grassland, loam | grassland | grassland, loam | long grassland | grassland, loam |
| Pha Din, Vietnam: evergreen needleleaf forest SGS=1, EGS=366 | evergreen needleleaf forest, loam | evergreen needleleaf temperate forest | evergreen needleleaf forest, clay | evergreen needleleaf forest | evergreen needleleaf forest, clay |
| Quabbin Reservoir/Harvard Forest tower, USA SGS=93, EGS=312 | temperate mixed forest, loam | deciduous broadleaf temperate forest | deciduous broadleaf forest, sandy loam | deciduous broadleaf forest | deciduous broadleaf forest, sandy loam |
| Nebraska, USA: crops (maize/soybean rotation) SGS=132/148, EGS=278/260 | crops (maize, soybean), loam | C3 crop | crops/grassland mosaic, silty clay loam | crops | crops (corn), silty clay loam |
| Huancayo, Peru: grassland SGS=1, EGS=366 | grassland, loam | grassland | grassland, loam | long grassland | grassland, loam |
| Mt. Kenya, Kenya: grassland, shrublands SGS=1, EGS=366 | grassland, loam | grassland | grassland, loam | long grassland | grassland, silty loam |




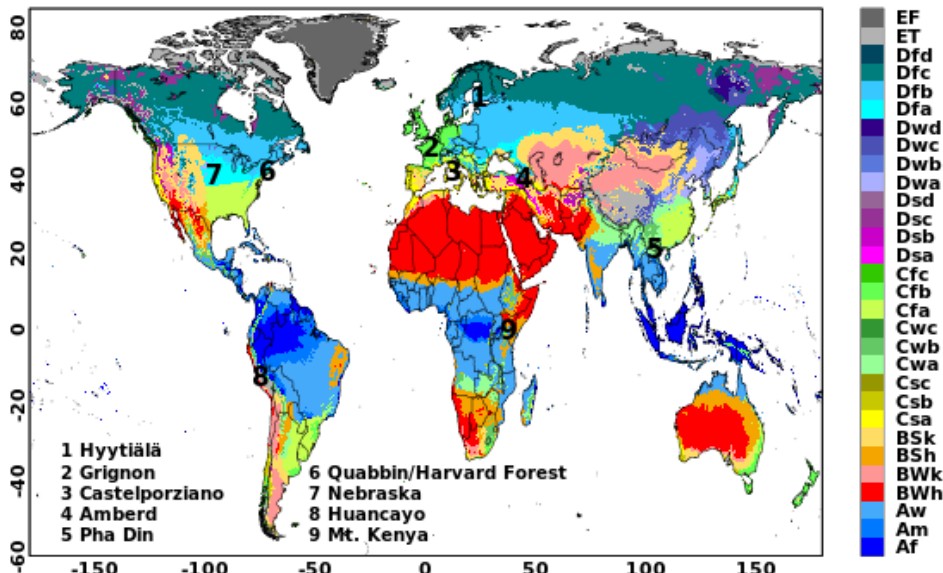


**Fig. 1: Locations of 9 selected sites on Köppen-Geiger climate classification map for 1991-2020 (source:**

**Beck et al., 2023). Table 1 specifies the classifications of these sites.**
**2.3 Stomatal deposition models and their key inputs**
Six widely used empirical/Jarvis and semi-empirical/Ball-Berry types of stomatal deposition models were
selected for this study. All of these used models can accommodate a variety of land cover/land use types and
provide estimates of stomatal deposition that can be output as both hourly- and season-long cumulative-stomatal
deposition metrics. The model details are described below.
(1) Empirical/Jarvis-type models - The ZHANG model modifying a predefined minimum stomatal
resistance for sunny and shaded leaves with environmental stress functions in Jarvis-Style (Zhang et al.,
2002; 2003; 2006), the Web-DO$_3$SE (i.e., a version of DO$_3$SE that is directly coupled to the TOAR
database) model modifying a predefined maximum stomatal conductance with phenology and
environmental stress functions depending on radiation (PAR), vapour pressure deficit (VPD) and soil
water (SM) (Emberson et al. 2000), the CMAQ_J model modifying a minimum stomatal resistance
with stress factors for PAR, air temperature (air T) and relative humidity (RH) at leaf surface, and root
zone soil moisture (Pleim and Ran, 2011), the MESSy model instead calculates the initial stomatal



conductance based on the PAR (instead of using a stress function) (Ganzeveld et al., 1995; Kerkweg et
al., 2006)
(2)  Semi-empirical/Ball-Berry - The CMAQ_P model using linear regression for C3 and C4 plants based
on $CO_2$ net assimilation (Ran et al,. 2017), the TEMIR model solves the coupled photosynthesis-
stomatal conductance system (Collatz et al., 1991; Farquhar et al., 1980) separately for sunlit and
shaded leaves (Tai et al., 2024; Sun et al., 2022) with distinction between C3 and C4 photosynthesis
(Collatz et al., 1992), the NOAH-GEM model involves additionally RH instead of VPD (Wu et al.,
2011; Niyogi et al., 2009).
All models follow the resistance scheme:

179                    +— Rcut

i.e.     Ra — Rb — + — Rstom

181                    + — Rinc + Rground

The land cover, growing season, and soil texture specifications used by the models are summarised in Table 2.
For crops, we used the GGCMI Phase 3 crop calendar (Jägermeyr et al., 2021a) which provides the planting date
and maturity day for 18 different crops at a 0.5° land grid cell resolution (Jägermeyr et al., 2021b). For forest
trees, we consider four various classes: evergreen-needleleaf (EN), evergreen-broadleaf (EB), deciduous
needleleaf (DN), and deciduous broadleaf (DB). For evergreen species, we assume a year-round growing season;
for deciduous species, we used the simple latitude function described in Hayes et al.(2017); and we consider a
year-round growing season for tropical species. The soil texture categories used by the models were obtained
from the reference studies in Table 1 and the site principal investigators. Table 3 provides the key formulas,
input data requirements and references for all models. Key total and stomatal deposition parameters for
empirical models ($g_{max}$) and semi-empirical models ($V_{Cmax}$) are described in Table 4, which gives a good
indication of the overall difference in the magnitude of stomatal deposition. The models' meteorological and $O_3$
inputs have been introduced in Section. 2.1.



**Table 3. Stomatal deposition models selected for site-scale modelling (list of symbols: A1 and Section S3 in the SI, *uses u(h), o3(h)=1, for US-Ne: u(h), o3(h)=0.3).**

| Model | Approach | Key Formulas | Key Input data | Reference |
|---|---|---|---|---|
| ZHANG | Empirical (Jarvis-style) | $R_s = 1/[G_s(PAR)f(T)f(D)f(\psi) \times D_i/D_v]$ <br><br> $G_s(PAR) = \dfrac{L_{sun}}{r_s(PAR_{sun})} + \dfrac{L_{shade}}{r_s(PAR_{shade})}$ <br><br> $r_s(PAR) = r_{s,min}(1 + b_{rs}/PAR)$ | LAI, LUC, Wspeed, ssrd, T2m, Tskin, RH | Zhang et al., 2002; 2003; 2006 |
| Noah-GEM | Semi-empirical, photosynthesis-based (Ball-Berry type) | $R_s = 1/[LAI(mA_n h_s P/C_s + b)]$ | LAI, LUC, Wspeed, ssrd, strd, T2m, Tskin, RH | Wu et al., 2011; Niyogi et al., 2009 |
| CMAQ_J | Empirical (Jarvis-style) | $R_s = r_{s,min} LAI/(f_{PAR}f_T f_{vpd}f_w)$ | LAI, Tair, PAR, ssrd, rn, RH SM | Pleim & Ran 2011 |
| CMAQ_P | Semi-empirical, photosynthesis-based | $R_s = 1/(m_g A_{net} e_s P_a/C_s e_i + g_0)$ | LAI, CO₂, Pa, u*, | Ran et al,. 2017 |





| | (Ball-Berry type) | | h_dis, z0, SM, Tsoil, wspeed, wdir, Soil texture, C3/C4 type, PAR, ssrd, rn, P_rate, sn, sd | |
|---|---|---|---|---|
| **TEMIR** | Semi-empirical, photosynthesis-based (Ball-Berry type) | $R_s = 1/\left[\left(\frac{L_{sun}}{r_b+r_{sun}} + \frac{L_{shade}}{r_b+r_{shade}}\right)\frac{D_i}{D_v}\right]$ <br><br> $r_s = 1/g_s = 1/\left[\alpha\left(\frac{mA_n\left(\frac{e_s}{e_{sat}}\right)}{\left(\frac{C_s}{P_{atm}}\right)} + b\right)\right]$ | LAI, LUC, u*, ssrd, T2m, Tskin, RH, SM | Tai et al., 2024; Sun et al., 2022 |
| MESSy | Empirical (Jarvis-style) | $R_s = [r_s(PAR, LAI)/f_T f_{vpd} f_w)] \times D_v/D_i$ <br><br> $r_s(PAR, LAI)$ <br> $= \dfrac{kc}{[\frac{b}{d\,PAR}\ln\left(\frac{d\exp(kLAI)+1}{d+1}\right) - \ln\left(\frac{d\exp(-kLAI)}{d+1}\right)]}$ | LAI, ssrd, RH, sw, Tir | Emmerichs et al., 2021; Kerkweg et al., 2006; Ganzeveld et al., 1998 |





| Web-DO₃SE | Empirical (Jarvis-style) | $r_s = g_{max}\ max\{(f_{min}, f_{temp}, f_{VPD}, f_{SWC})\} \times f_{phen} \times f_{light}$ | Tair, VPD wspeed, P, Pa, O₃, Gr | Emberson et al., 2000; Bueker et al., 2012; Simpson et al., 2011; Guaita et al., 2023a |
|---|---|---|---|---|



**Table 4 Model parameter $V_{Cmax}$ [in µmol $CO_2$ m⁻² s⁻¹ ] and $g_{max}$ [$O_3$ in cm s⁻¹ ] by land cover/land use type.**
**Note that the values presented in the table were recalculated from the original respective rsmin values for**
**H₂O (s m⁻¹) in ZHANG, MESSy, and CMAQ_J, and Vc_max values for O₃ (mol O₃ m⁻²s⁻¹) in Web-D O₃SE .**

| Parameter | Web-DO₃SE | ZHANG | CMAQ_J | TEMIR | NOAH-GEM | CMAQ_p |
|---|---|---|---|---|---|---|
| **G_max or V_Cmax** | g_max [cm s⁻¹] | g_max [cm s⁻¹] | g_max [cm s⁻¹] | $V_{Cmax}^{+}$ [µmol $CO_2$ m⁻² s⁻¹] | $V_{Cmax}$ [µmol $CO_2$ m⁻² s⁻¹] | $V_{Cmax}$ [µmol $CO_2$ m⁻² s⁻¹] |
| Forests | 0.44 (EN) 0.49 (EB) 0.55 (DB)  Zhang et al., 2003 | 0.25 (EN) 0.42 (EB) 0.42 (DB) | 0.36 (EN), 0.53 (EB), 0.32 (DB),  Pleim & Ran, 2011 | 60.1 (EN) 59.0 (EB) 55.4 (DB),  (Oleson et al, 2013; NCAR Technical | 57.6 (EN) 96 (EB) 96 (DB)  Niyogi et al., 2009; JAMC | 57.6 (EN, Slevin et al 2015), 49.2 (EB, Medi. forest, (EB_tr+EB _te)/2,Oliv |



| | | | | notes) | | er et al., 2022), 55.4 (DB, CLM4.5, Kattge 2009) |
|---|---|---|---|---|---|---|
| Crops | 1.1 (wheat) 0.74 (maize) 0.73 (soybean) | 0.53 | 0.91 | 96.7 | 76.8 | 96.7 (CLM4.5) |
| Grasses | 0.66 | 0.64 | 0.64 | 75.1 | 28.8 | 75.1 (CLM4.5) |


$POD_y$ is calculated in post-processing, according to the guidelines in UNECE LRTAP (2017):.
$$POD_y = \sum_{i=1}^{n}[fst,sun_i - y] * \left(\frac{3600}{10^6}\right) \quad \text{for } fst,sun_i \geq y\,nmol\,m^{-2}\,PLAS^{-1}$$
Where $fst,sun_i$ is the hourly mean $O_3$ flux in nmol $O_3$ m$^{-2}$ PLA s$^{-1}$ at sunlit leaves, $y$ is a species-dependent
threshold (crops: 6 nmol $O_3$ m² s$^{-1}$, grassland and forests:1 nmol $O_3$ m$^{-2}$s$^{-1}$; UNECE LRTAP (2017) and i is the
number of daylight hours (when ssrd > 50 W m$^{-2}$) within the accumulation period (growing season). The term
(3600/10$^6$) converts from nmol m$^{-2}$ PLA s$^{-1}$ to mmol $O_3$ m$^{-2}$ PLA. $fst,sun$ is calculated by:
$$f_{st,sun} = c\,\frac{(z) * g_{st} * r_c}{r_b + r_c}$$
Where c(z) is the $O_3$ concentration at in nmol m$^{-3}$ (calculated from ppb by multiplying by P/RT where P is the
atmospheric pressure (Pa) and T is the air temperature (K)



, R is the universal gas constant of 8.31447 J mol$^{-1}$ K$^{-1}$ and T is the assumed standard air temperature (293 K).
The leaf surface resistance ($r_c$) is given by $r_c = 1/(g_{st}+g_{ext})$ where $g_{ext}$ is the inverse of cuticular resistance.
. The leaf boundary resistance is calculated by:
$$r_b = 1.3 * 150 * \sqrt{\frac{L}{u(h)}}$$

Where factor 1.3 accounts for the differences in diffusivity between heat and $O_3$, $L$ is the crosswind leaf
dimension (i.e. leaf width in m) and $u(h)$ is the wind speed at the top of the canopy.
**2.4 Description of stomatal deposition model simulations**
The result section aims at identifying trends in stomatal deposition models among different land cover types
including grass, crops and forests using four model experiments as follows.
**In experiment 1**, the different models are driven by the $O_3$ and meteorological data from ERA5. We analysed
the simulated deposition velocity ($V_d$) split into stomatal and non-stomatal fractions, canopy ($G_{st}$) and sunlit
($G_{sun}$) stomatal conductance.
To include observational constraints, in **experiment 2,** the TEMIR, ZHANG, NOAH, MESSy and CMAQ
models were run with data obtained from the FLUXNET database (available for three sites, see Table 1), and the
simulated $G_{st}$ was evaluated with observation-derived values, inferred $G_{st}$, of SynFlux. Spearman correlation was
applied for the model evaluation, as it can be applied to any datasets including non-parametric and non-linear
ones. The US-Ha1 and FI-Hyy sites were considered for the model evaluation due to the availability of SynFlux
data at these sites
A sensitivity analysis (**experiment 3**) was performed by driving a set of models with synthetic input data in the
following steps: i. $O_3$ input was perturbed by +/- 40% (Sofen et al. 2016). ii. soil water content was perturbed by
+/- 30 % (Li et al., 2020). iii. absolute humidity was perturbed by +/- 30%, soil and air temperatures were
perturbed by +/-3, independently, iv. the growing season, which was mostly approximated by LAI, was shifted
by 14 days forward and backward in time. In set (iii) and (iv), relative humidity was calculated from absolute





humidity and temperature after their perturbation. In both cases, absolute humidity was capped at the saturation
vapour pressure at the corresponding temperature.
Finally, for **experiment 4,** $g_{max}$ and $V_{Cmax}$ of the models were varied by +-20 %, based on previous estimates of
plant traits dependent uncertainty (e.g., Walker et al., 2017; UNECE LRTAP, 2017).
**3. Results**
**3.1 General characteristics of simulated total deposition velocity and stomatal contribution**
The split of total $O_3$ deposition between different pathways, $G_{st}$, $G_{cut}$, $G_{ground}$, simulated by the 7 models is shown
for each of the 9 sites in Figure 2 (corresponding data are presented in Table S9). This analysis allows us to
briefly assess the overall efficacy of the model's ability to simulate deposition velocity $V_d$ (by comparisons with
previously published values; more complete assessments of model's ability for some of these sites can be found
in Clifton et al., 2023) and to compare the importance of the stomatal deposition pathway between models for
different land cover types and across different seasons.
Observations of $V_d$ have only been made at a handful of sites i.e. Hyytiälä, Finland; Castelporziano, Italy;
Grignion, France and Harvard Forest, US (close to our Quabbin site in terms of proximity, land cover type and
climate). Overall, the models capture $V_d$ at these sites compared to observed values reported in previous studies.
Namely, the observed seasonal cycle in $V_d$ at Hyytiälä, Finland (needleleaf forest), with lows of ~0.1 cm s$^{-1}$
between January and April and highs of 0.4 cm s$^{-1}$ between June to September, averaged over 10 years of
measurements from Clifton et al. (2023) and Visser et al. (2021) are captured by most models except of MESSy
and TEMIR, which reach $V_d$ values of 0.8 cm s$^{-1}$ during the summer. Similarly, the strong seasonal cycle in $V_d$
at Quabbin, US (temperate mixed forest), ranging from around 0.2 cm s$^{-1}$ between January and April up to 0.5
cm s$^{-1}$ from June to September in Clifton et al. (2023) is captured by all models. Observed $V_d$ at Castelporziano,
Italy (evergreen broadleaf forest) shows relatively constant values throughout the year, commonly between 0.4
and 0.8 cm s$^{-1}$ averaged over a 2-year period (Savi & Fares, 2014). The study by Stella et al. (2011) reports $V_d$
measurements of 0.63 cm s$^{-1}$ (on average) at Grignion (France). At the other sites, no $O_3$ dry deposition
measurement exists and thus we report the observed ranges for the land cover type (and possibly the matching
climate). Over grassland, Silva and Heald (2018, and references therein) show a mean of 11 measurements of
daytime $V_d$ values (~0.4 cm s$^{-1}$) in agreement with our models. Measurements exist at soybeans and maize crops





which indicate $V_d$ values of 0.7 (Meyers et al., 1995) and 0.4 - 0.6 cm s$^{-1}$ (Stella et al., 2011), respectively. Thus,
the models seem to estimate too low deposition at soybeans.
In terms of deposition pathways, for all sites and models, stomatal deposition consistently ranks as the most
important pathway in the summer, whereas in winter and, for some models, in the fall $G_{st}$ decreases to zero to
very low at sites with seasonal variation in vegetation coverage. The importance of the pathway varies with land
cover type and season. The highest stomatal contribution of 90 % (NOAH model) is shown at the Amberd site.
Among the different land cover types, the highest average stomatal contribution to deposition during the summer
is estimated across grass (67 %), followed by crops (65 %) and forests (59 %). The seasonal importance of
stomatal contribution is not seen for the tropical sites as the year- round growing season means that stomatal
conductance is driven by solar radiation which is constant throughout the year (e.g. Hardacre et al., 2015).
Previous studies involving measurements and partitioning approaches (Horvath et al., 2018, Meszaros et al.,
2009) indicate that the non-stomatal $O_3$ deposition pathways (i.e., $G_{ground}$ and $G_{cut}$) are very strong (in some
cases, dominant over $G_{st}$) at short vegetation such as the grasslands. Despite there are multiple factors such as
wind speed, solar radiation, LAI, etc., that control the relative contributions of the three deposition branches, $G_{st}$
is the dominant pathway at the three grassland sites of the current study (Amberd, Mt Kenya, and Peru). At the
Amberd and Perus sites, $G_{cut}$ and $G_{ground}$ are low due to lower wind speeds (e.g., at the Peru Site in the Summer
season, the mean wind speed was 1.0 cm s$^{-1}$ and the Gut and $G_{ground}$ contributions in the TEMIR model were
21 % and 12 %, respectively; Table S3).
In contrast, at the Mt Kenya site, $G_{st}$ overcomes the $G_{cut}$ and $G_{ground}$ due to higher solar radiation at this site
(annual mean is 246 W m$^{-2}$, Table S2). Therefore, it can be inferred that the $O_3$ deposition pathway depends on
not only the land cover type but also meteorological drivers. Among the models, Web-DO$_3$SE estimated the
lowest stomatal contribution at grass (Fig. 2) most likely due to its parallel pathways to cuticle, soil and stomata,
with the former scaled by LAI with a constant cuticular deposition of 2500 s m$^{-1}$. Such differences in model
structures likely led to the wide-ranging partitioning. For example, for the Quabbin site (summer), all models
simulate $G_{cut}$ ranging from 15-65 %, $G_{ground}$ from 2-19 % and $G_{st}$ from 33-66 % despite their agreement on the
overall $V_d$ values (total bar). Models agree better in the partitioning of $O_3$ dry deposition to crops with summer
stomatal fraction contributions ranging between 46-73 %, 37-73 % and 51-81 % for US-Ne3 Maize, US-Ne3
soybeans and FR-Gri (rapeseed and wheat). Most models estimate non-stomatal deposition equal to or larger
than the stomatal contribution to deposition outside of the tropics in winter and fall, and to some extent in spring.
This again emphasises the importance of the stomatal contribution to the seasonal cycle of total deposition as



also found in Clifton et al. (2023). Similarly, as seen at grasslands, Web-DO$_3$SE (Fig. 2, Table S3) accounts for
the highest non-stomatal deposition at crop sites.
Across all forest sites, models show significant cuticular uptake throughout the year ranging between 11 % and
94 % contribution. At FI-Hyy, $G_{cut}$ averages ~50 % across all seasons and all models with higher estimates of
~55 % by the TEMIR model due to the higher wind speed at FI-Hyy (annual mean wind speed is 3.2 m s$^{-1}$; Table
S2) favoring cuticular deposition as suggested by Rannik et al. (2012). At IT-Cpz, our models estimate on average
around 43 % (20-80 %) to be non-stomatal deposition, close to the previously reported ranges (Gerosa et al. 2005,
Fares et al. 2012, Fares et al. 2014), which were up to 57 % from non-stomatal deposition and 30-60 % from
stomatal uptake. A similar partitioning (59 % $G_{st}$, 33 % $G_{cut}$, 5 % $G_{ground}$ model average in summer) is seen at
PhaDin.










**Fig. 2 Seasonal mean effective conductances of the cuticular ($G_{cut}$), ground ($G_{ground}$), and stomatal ($G_{st}$) deposition pathways of $O_3$ across various models and sites (Exp#1).**

All models except Web-DO$_3$SE were compared on a seasonal and hourly basis with the SynFlux $G_{st}$ data for US-Ha1 and FI-Hyy sites (Figures S2, S3). CMAQ_J, NOAH, TEMIR, and ZHANG show reasonable agreement at the Quabbin forest (US-Ha1) whereas CMAQ_P and MESSy show quite significant overestimates at both FI-Hyy and Harvard Forest (Table S4) and CMAQ_J overestimates at FI-Hyy only. Note that NOAH and ZHANG show significant underestimates at FI-Hyy while agreeing well with SynFlux at Harvard Forest (Quabbin). The underestimates by the ZHANG model are consistent with the results from a similar comparison for Yellowstone National Park in the US by Mao et al. (2024). Compared to Harvard Forest, FI-Hyy is the most humid and cloudy with the lowest solar radiation flux, and these conditions likely contribute to the underestimates by the NOAH and the ZHANG model as identified by Mao et al. (2024). The differences between modelled and SynFlux $G_{st}$ do not seem to be associated with the model types, i.e. empirical or photosynthesis-based models.

The correlation of the diurnal cycle of $G_{st}$ calculated by the models with the inferred $G_{st}$ by SynFlux for US-Ha1 and FI-Hyy (Fig. S4) confirms that models generally capture the temporal patterns of $G_{st}$ of at least these two different forest types and climates (FI-Hyy: EN, temperate, subarctic; Quabbin: DB, moist temperate). The best Spearman correlations are found at FI-Hyy and range between 0.73 by the MESSy model and 0.85 by the TEMIR model. Overall lower correlations are found at the Quabbin site ranging from 0.65 for the NOAH and MESSy models to 0.82 for the CMAQ_P model. This poorer correlation suggests that additional water stress may limit stomatal conductance at Quabbin, which the models do not capture, compared to FI-Hyy. Notably, a similar range of correlation coefficients (0.61 – 0.93) was found when modelled $G_{st}$ values obtained using the TOAR input data were compared with SynFlux $G_{st}$. As SynFlux data were generated using FLUXNET measurement data, this result corroborates the validity of using the TOAR database as input to Web-DO$_3$SE, developed as a service website to aid in risk assessment of $O_3$ damage to European vegetation.

To identify the key drivers of the $G_{st}$ model schemes among different land cover types and climate conditions, we also compare estimates of $G_{st}$ between models for all sites and analyse the similarity of $G_{st}$ diurnal cycles in empirical and photosynthesis models. The average diurnal variations of stomatal conductance (Gst) of O3 at the





9 sites for each season are shown in Figure 3. This also helps interpret the modelled stomatal conductance of
sunlit leaves ($G_{sun}$) shown in Fig 4. Across all models, the diurnal mean $G_{st}$ (Fig. 3) varied from 0.15 cm s$^{-1}$
(Quabbin) to 0.50 cm s$^{-1}$ (Mt. Kenya). In the TEMIR and the ZHANG model, roughly 50% of $G_{st}$ occurs at the
sunlit part of the leaves. Web-DO$_3$SE and CMAQ_P $G_{sun}$ contribute 30 % on average (Fig. 4). At mid-to-high
latitudes, the model spread is limited to the summer season, whereas at tropical sites, it is similar throughout the
year. During the day, models show a spread of 1.2 cm s$^{-1}$ in $G_{st}$ at the forest and grassland sites during the
summer while their predictions agree most at the crop sites (throughout the year) with a maximum of 1.0 cm s$^{-1}$.
This is due to the flux response relationship which has a more sensitive response (steeper slope for most crops)
due to a higher threshold (see Table 5 for the equations describing the steepness of the change). Results among
the same model type differed significantly while different model types could produce similar results at the same
location. For the sites with distinct seasonal variations, model differences were the largest in summer.
In comparison, TEMIR and ZHANG, photosynthesis-based and Jarvis-style, respectively, are both governed
mainly by solar radiation (see higher $G_{sun}$ in Fig. 4), showing close agreement, except in summer, at the forest
sites (ZHANG values are very low). Only these two show a midday depression in $G_{sun}$ at the peak of solar
radiation at Mt Kenya (the site with the highest radiation). The ZHANG model also estimated this feature for
$G_{sun}$ and $G_{st}$ at other grassland sites (Fig. 3 and 4). This feature could be due to the day length (seasonality)
scaling of $V_{Cmax}$ in TEMIR, causing $G_{st}$ to increase significantly during summer at higher latitude sites. In
contrast, at lower latitude sites (Mt Kenya and Huancayo, Peru), the seasonal variation in day length is smaller
and subsequently smaller seasonality in $V_{Cmax}$ and $G_{st}$. The TEMIR and the CMAQ_P models, both
photosynthesis-based, estimate very similar $G_{sun}$ values (Fig. 4) at PhaDin (fall, winter), IT-Cpz (spring,
summer) and FI-Hyy (summer) whereas the $G_{st}$ estimates show significant differences. The opposite occurs at
Quabbin where the $G_{sun}$ values of the two models differ much more than the $G_{st}$ estimates. These results
illustrate that the different fractionations between shaded and sunlit leaves could mainly contribute to the model
spread in stomatal conductance.
Further examination of individual models' features can shed light on the causes of model/site differences in $G_{st}$.
The MESSy $G_{st}$ value is strongly governed by LAI followed by soil moisture, and in all other respects MESSy
treats different land cover types the same. Therefore, MESSy simulates the highest $G_{st}$ values at PhaDin,
Grignion and Mt. Kenya with LAI values of 6.9, 4.3 and 4.2 m² m$^{-2}$, respectively (Table 1). In contrast to
PhaDin, the high LAI site IT-Cpz (6.9 m² m$^{-2}$) experiences significant water stress during summer. This is only
captured by MESSy and NOAH indicating higher sensitivity to water stress. During the day, an evident midday



depression of $G_{st}$ due to hot weather and water shortage is seen accompanied by a peak in the early morning
evident from NOAH, same as has been observed in Mediterranean ecosystems (e.g. Gerosa et al. 2005). The
NOAH model accounts for the direct effect of relative humidity on $G_{st}$ (see model description in the supplement)
and subsequently modelled a depression in $G_{st}$ at the daily onset (8 am). This variation explains the $G_{st}$ peak at
IT-Cpz and Quabbin, which are especially in the summer the two driest among all sites. Due to the dry
conditions at the Quabbin site, low soil water and relative humidity, most models, except NOAH, simulate the
lowest summer daily mean $G_{st}$ values among all sites. The high estimate by the NOAH model can be explained
by the highest $V_{Cmax}$ value among the photosynthesis models (Table 4). The high $g_{max}$ value of 0.55 cm s$^{-1}$ used
in Web-DO$_3$SE leading to large estimates is largely dampened by strong soil moisture stress at IT-Cpz (Table
S2). Similarly, Web-DO$_3$SE estimates the lowest $G_{st}$ (among the models) values at the Peru site (grassland) due
to a strong limitation by the $f_{temp}$ function on stomatal conductance suggesting that the minimum temperature for
stomatal opening at 12 °C is too low for these cool temperate conditions. The ZHANG estimates are generally
governed by $g_{max}$, explaining the highest and lowest $G_{st}$ values of all models simulated with the ZHANG model
at grassland and forest sites, respectively. The CMAQ_J model has the lowest $g_{max}$ values, but it is strongly
impacted by soil moisture. The additional dependence of the ZHANG model on solar radiation is reflected in
higher $G_{sun}$ relative to $G_{st}$ (Fig. 3 and 4). TEMIR also simulates the smallest spread of $G_{st}$ among the 3 grassland
sites (Ambred, MKenya, Peru), as temperature acclimation of photosynthesis (Kattage and Knorr, 2007) is
implemented. The different temperatures among the 3 sites have smaller effects on photosynthetic capacity and
$G_{st}$ than other models. Despite explicitly considering soil water stress, TEMIR does not capture the impacts of
water stress on $G_{st}$ in IT-Cpz and Quabbin in the summer, as the equivalent soil moisture threshold to trigger soil
water stress at IT-Cpz and Quabbin is very low (<0.1 m³ m$^{-3}$). Both versions of CMAQ respond very strongly to
soil moisture which may not be accurate for each site. The differences between CMAQ-J and CMAQ-P are
greatest at the sites with the greatest LAI, such as IT-Cpz and PhaDin.







**Fig. 3 Multi-year diurnal cycle of growing season $G_{st}$ from models at 9 different sites. Four topmost panels are the forest sites, three panels in the middle are grass sites, and three lowermost panels are crop sites. Open circles indicate diurnal $O_3$ variations**







**Fig 4 Leaf level sunlit stomatal conductance ($G_{sun}$) from the 3 two-leaf models (CMAQ_P, TEMIR, and**
**ZHANG) at 9 different sites. Four topmost panels are the forest sites, three panels in the middle are grass**
**sites, and two lowermost panels are crop sites (US-Ne3-S=soybeans, US-Ne3-M=maize). Open circles**
**indicate diurnal $O_3$ variations**
The difference between total and sunlit stomatal flux is examined, and trends of stomatal sunlit flux are
characterized by different land cover types and climate conditions. Figures 5 and 6 show the (SRAD>50 W$m^{-2}$)
stomatal $O_3$ flux ($F_{st}$) and stomatal, sunlit $O_3$ flux ($F_{st,sun}$) for different models per season at 9 sites representing
forest (top), grass (middle), crops (bottom). Thereby, we consider whether $G_{st}$ and $O_3$ concentration co-variate at
diurnal and seasonal timescales. Across all land cover types, a large range of $F_{st}$ (0.05-2 ppb m s$^{-1}$, Fig. 5) is
estimated, usually highest in spring and summer and lowest in winter. The largest median of $F_{st}$ is found at
Amberd (0.75 ppb m s$^{-1}$; ZHANG, summer), followed by IT-Cpz (0.60 ppb m s$^{-1}$; NOAH, spring), and FR-Gri
(0.60 ppb m s$^{-1}$; MESSy and NOAH, summer) owing to both higher $G_{st}$ and $O_3$ concentrations at the respective
sites (Fig. 3). Consequently, no general trend can be identified among the sites, i.e flux estimates can differ
within one land cover type. Namely, the two crop sites show very different $F_{st}$ estimates (Fig. 5) since they have
the most different $O_3$ levels across one land cover type. While the FR-Gri site is exposed to an annual mean $O_3$
of 45 ppb (Table S1) as the lowest $O_3$ level of 25 ppb among all sites. The same applies for the diurnal variation
of $O_3$ causing either a high (FR-Gri) or a low range (US-Ne3) of flux estimates among all models (in summer
and spring). The difference is less apparent in the $F_{st,sun}$ estimates (Fig. 6) which point to the sensitivity of the
two leaves to $O_3$ concentration. Similarly, as seen for the stomatal conductance, three of four models show a
very good agreement of $F_{st}$ and $F_{st,sun}$ among each other. In terms of seasonality, models agree also generally
well among the grassland sites. Among those (and all land cover types), the maximum annual median $F_{st,sun}$ was
estimated for Amberd attributed to the high daytime (7 am - 7 pm) annual $O_3$ concentrations (49.3 ppb, Table
S1). The most different $F_{st,sun}$ (and $F_{st,sun}$) values are found between the ZHANG (highest) and Web-DO$_3$SE
model (lowest) due to the difference in $G_{sun}$ (Fig. 4). Web-DO$_3$SE disagrees the most with the other models
and predicts very small fluxes at the Peru site following the small $G_{st}$ and $G_{sun}$ values (Fig. 3 and 4).
Among forest sites, spring $F_{st,sun}$ values are comparably high as summer fluxes following the seasonal variation
of $G_{sun}$ (Fig. 6, outside the tropics). The highest spring estimates at PhaDin and Quabbin (forests) are linked to
the site-specific yearly $O_3$ maximum in this season (Fig. 3). The flux seasonal maximum is more pronounced in
all four models (ZHANG, CMAQ_P, TEMIR) when the $O_3$ concentration variation during the year is larger at
the respective site. The highest $F_{st,sun}$ (0.1 ppb m s$^{-1}$) is estimated by TEMIR at PhaDin (spring) reflecting the



high $G_{sun}$ estimate. In contrast, when considering the total $F_{st}$, CMAQ_P shows the highest estimate (Fig. 5)
which indicates that TEMIR uses a higher sunlit fraction than CMAQ_P as it has been shown for stomatal
conductance (Fig. 3 and 4). The difference is most apparent at high LAI sites (PhaDin, IT-Cpz, FR-Gri). The
lowest estimates of $F_{st,sun}$ (and a very small spread) at the forest sites are shown by the ZHANG model as it has
been explained for $G_{st}$ and $G_{sun}$. Overall, CMAQ_P has the lowest spread among the models which was also
found in the multi-model comparison study by Clifton et al. (2023).

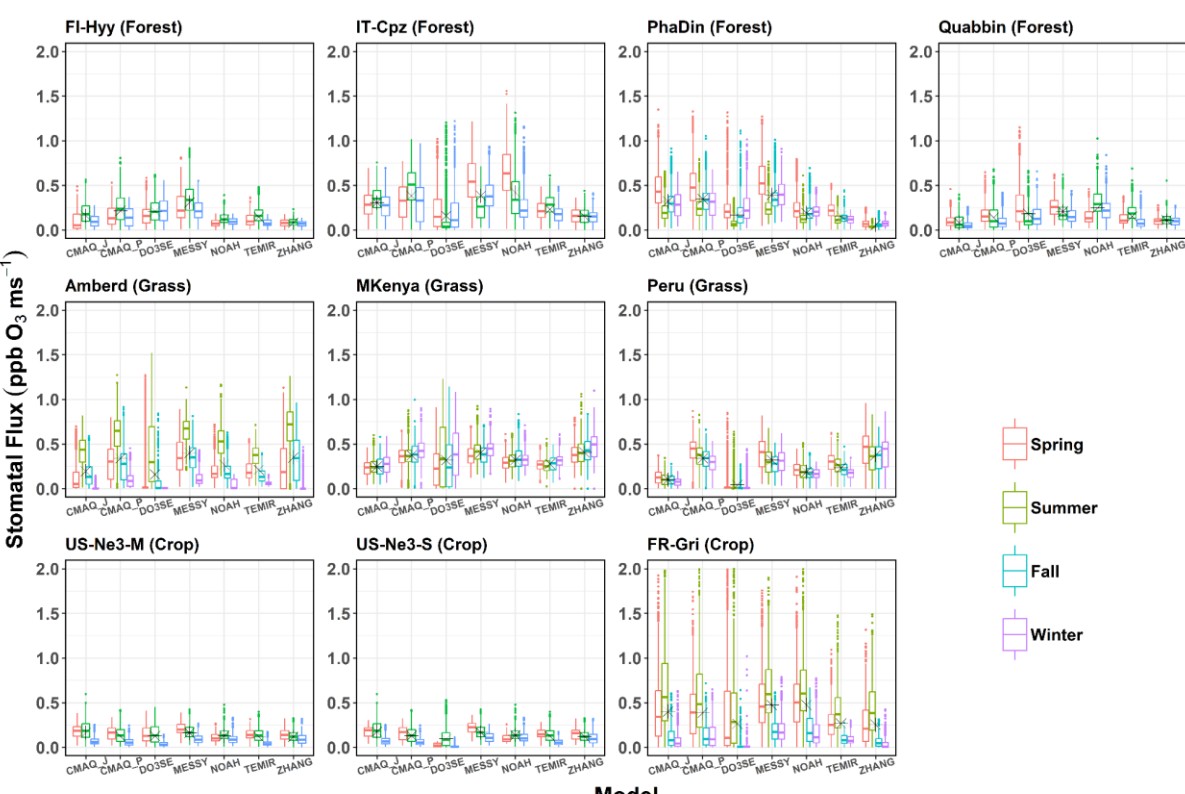


**Fig 5: Boxplots of seasonal mean canopy-level total stomatal O$_3$ flux (ppb ms$^{-1}$) for different models at the different 9 sites (data represent SRAD > 50 W m$^{-2}$ and the growing period).**





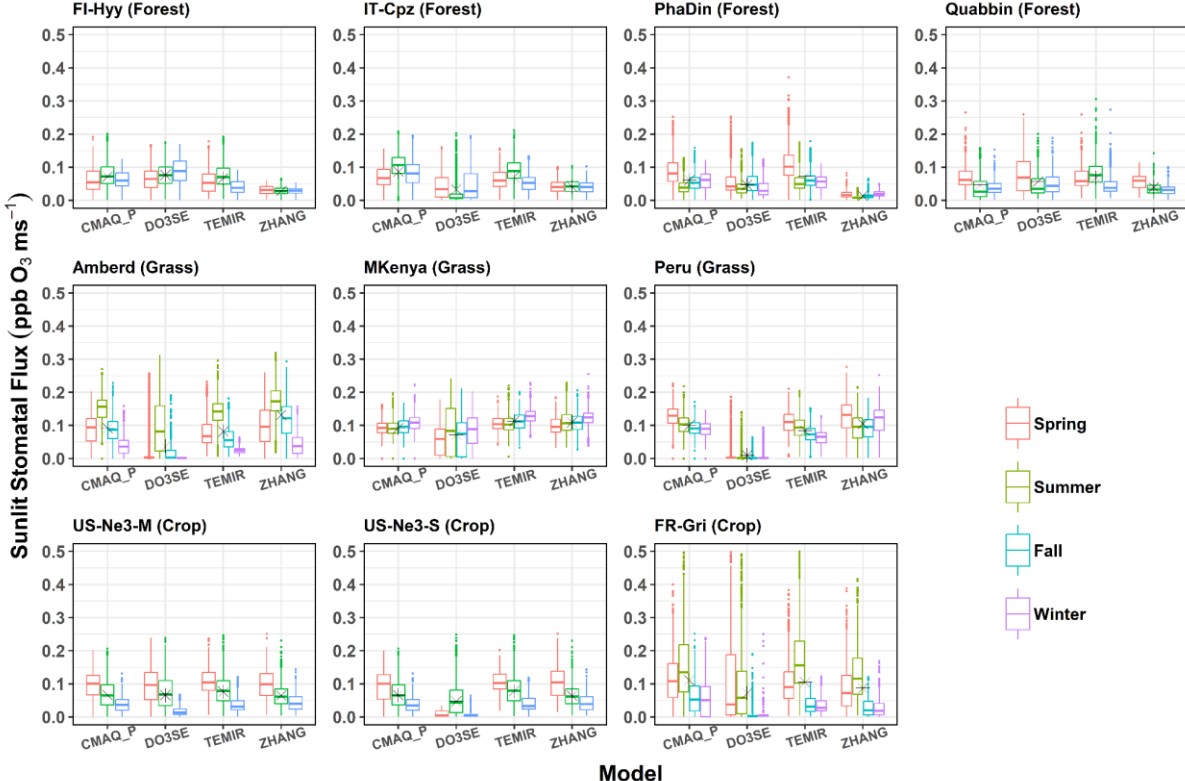

**Fig 6: Boxplots of seasonal mean leaf-level sunlit stomatal O$_3$ flux (ppb ms-1) for different models at the different 9 sites (data represent SRAD > 50 W m$^{-2}$ and the growing period).**

### 3.2 Vegetation impact and variation with key input data

This section presents the POD$_y$ calculated from the O$_3$ deposition by different models at 9 different stations to identify trends and patterns of POD$_y$ among land cover types and climates (Fig. 7, corresponding data in Table S9). By driving the models with changed input data of O$_3$, soil moisture, temperature, relative humidity, growing season (Fig. 8) and with changed Vc$_{max}$/g$_{max}$ parameter (Fig. 9) we explore the sensitivity of the POD$_y$ estimates. As shown in the previous analysis, the largest O$_3$ uptake and thus the highest POD$_y$ of 28 mmol O$_3$ m$^{-2}$ (on average among all models) is estimated over grassland sites (compared to forest and crops) (Fig. 7). POD$_1$ increases linearly with time for evergreen grasslands whereas Mt. Kenya shows the fastest accumulation (due to



the highest $F_{st}$ in spring and summer). Three of the four models lie in a range of 5 mmol $O_3$ m$^{-2}$ whereas Web-
DO$_3$SE predicts a maximum POD$_y$ of 10 mmol $O_3$ m$^{-2}$ at all grassland sites. Only at the Peru site, these low
values can be reasoned by the significantly lower Gssun and $F_{st,sun}$ (compared to other models).
For forests, our modelled ensemble POD$_1$ median and maximum values (ranging between 8 and 25 mmol $O_3$ m$^{-}$
$^2$) are similar in scale to values estimated across broad geographical regions by other studies. Karlsson et al.
estimated POD$_1$ values across Europe with the highest values in mid-latitude Europe for coniferous (15 to 20
mmol $O_3$ m$^{-2}$) and broadleaf (22 to 28 mmol $O_3$ m$^{-2}$) forests. However, the ZHANG and the Web-DO$_3$SE model
are estimated to be significantly lower POD$_1$ than CMAQ_P and TEMIR at each site. These estimates average to
16 mmol $O_3$ m$^{-2}$. There is no obvious pattern to which models tend to estimate higher or lower POD$_1$ values, but
these estimates are generally consistent with $G_{sun}$ (Fig. 4) and $F_{st,sun}$ (Fig. 6) model estimates explained by
particular model constructs or parameterisations. For instance, the ZHANG model estimates low stomatal
deposition and thus also POD$_y$ over all forests. Web-DO$_3$SE saw a low $O_3$ uptake only due to the site conditions
at IT-Cpz.
For crops, the model estimates of POD$_6$ are a little more consistent, with modelled differences within sites only
varying between ~ 3 and 11 mmol $O_3$ m$^{-2}$, however, this could in part be due to the overall lower POD$_6$ values
due to the use of the higher y threshold. Median model ensemble values range between ~7 and 12 mmol $O_3$ m$^{-2}$
across sites. POD$_6$ for staple crops has been estimated in other studies across Europe and globally.  A European
study (Schucht et al., 2020) on wheat found POD$_6$ values up to ~ 4 mmol $O_3$ m$^{-2}$ suggesting that our POD$_6$
values for the FR-Gri site tend to be too high. Feng et al. (2012) estimated maximum POD$_6$ values of up to 8
mmol $O_3$ m$^{-2}$ for winter wheat in China though these higher values are likely driven by higher ozone
concentrations. Similarly, Wang et al. (2022) also found POD$_6$ values for maize of up to 8 mmol $O_3$ m$^{-2}$. Our
models give the largest range in POD$_6$ estimates for soybeans at the US-Ne3 site (0 to 11 mmol $O_3$ m$^{-2}$). A key
determinant of the range in POD$_y$ simulated by our models, and also with estimates provided in the literature, is
the value chosen for $g_{max}$ (or $V_{Cmax}$ depending on the model construct). For example, the multiplicative $g_{sto}$
models used to derive flux-response relationships (see Table 5) use $g_{max}$ values of 450, 126 and 301 mmol $O_3$ m$^{-2}$
s$^{-1}$ for wheat, maize and soybeans (UNECE LRTAP, 2017; Peng et al., 2019 and Zhang et al., 2017). By
contrast, our modelling uses a variety of $g_{max}$ values, for example, the Web-DO$_3$SE model uses 450, 305 and 300
mmol $O_3$ m$^{-2}$ s$^{-1}$ for wheat, maize and soybeans. A further consideration in parameter selection are local
conditions, a study by Stella et al., (2013) found a $g_{max}$ value of 296 mmol $O_3$ m$^{-2}$ s$^{-1}$ was most appropriate to
describe wheat $g_{sto}$ at the FR-Gri site. This variation highlights the importance of selecting appropriate model




parameterisation for conditions, as well as consistency of parameterisation with models used to develop flux
response relationships.

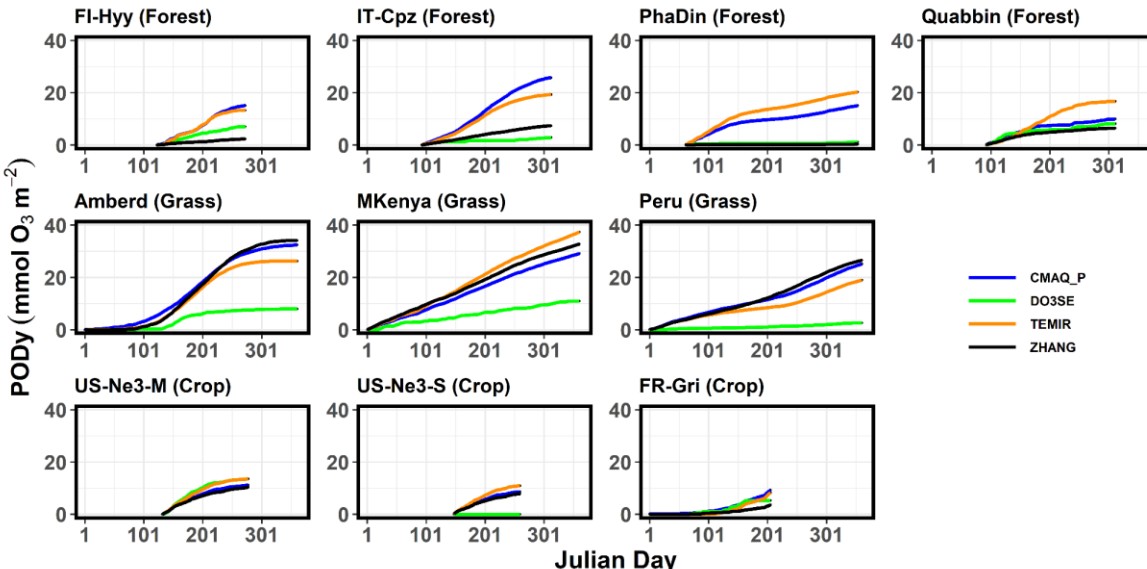

**Fig 7: Evolution of POD$_y$ (mmol O$_3$ m-2) through the growing seasons at various sites.**

From the sensitivity analysis, we found that all models show sensitivity of POD$_y$ to changes in O$_3$, specific
humidity, and temperature with varying degrees over different land cover types possibly due to different
prescribed values such as the temperature threshold (Fig. 8, corresponding absolute values in Table S10).
Especially, the POD$_y$ at all sites is most significantly changed when modifying the O$_3$ concentration by +-40 %
(Table S11). Crop is the most sensitive land cover to O$_3$ changes across the different models (8.5 mmol m$^{-2}$;
76 % POD$_y$ change with respect to the base run), followed by forest (10.0 mmol O$_3$ m$^{-2}$; 59.3%) and grass (14.9
mmol O$_3$ m$^{-2}$; 56.1%) which is due to the plant physiognomy (Grulke and Heald et al. ,2020). In a relative sense,
the average response change in POD$_y$ to a 40% change in O$_3$ concentrations is the greatest in ZHANG (+9.2
mmol/m2, corresponding to a 68.1 % POD$_y$ change with respect to the base run), followed by CMAQ_P and
TEMIR (12 and 11.9 mmol O$_3$ m$^{-2}$ ;64.8 % and 63.5 %), and then by Web-DO$_3$SE (11.4 mmol O$_3$ m$^{-2}$; 53.0 %).



Also, the $POD_y$ estimate seems to be sensitive to humidity (Q) changes (+-30%) among all models. At forest, the
$POD_y$ estimates appear to be the most sensitive (4.6 mmol/m2; 27.3%), followed by crops (2.9 mmol $O_3$ m$^{-2}$;
25.9%) and grass (4.6 mmol $O_3$ m$^{-2}$; 17.3 %). The response is the greatest in TEMIR and CMAQ (between 5.7
and 6.7 mmol m$^{-2}$; 30.7-35.8 %), while it is much smaller for ZHANG (usually close to zero on average). The
most non-linear response was shown by Web-DO$_3$SE at IT-Cpz which estimated a 5 times higher $POD_y$ response
to increasing humidity than to a humidity decrease pointing towards the strong dryness at this site limiting   If
temperature is changed by +-3 K the highest sensitivity was found at crops on average (2.7 mmol $O_3$ m$^{-2}$;
24.1%), followed by grass (4.6 mmol $O_3$ m$^{-2}$; 17.2 %) and forest (1.6 mmol m$^{-2}$; 9.5%). The responses unevenly
vary in sign depending on the model because the temperature change depends on the optimal temperature at the
specific sites. Namely most models estimate a $POD_y$ decrease when increasing temperature (Fig. 5). As
described in Hayes et al. (2019), a temperature increase is seen in southern countries where temperature could
limit stomatal uptake since temperature is already close to the optimum in normal conditions. From our
sensitivity analysis, temperature impacts on $POD_y$ are noticeable only for a few sites (e.g., Ambered, Mt. Kenya,
and Peru) and models's response to $POD_y$ change were different due to different thresholds used for the
temperature stress factors to stomatal conductance. The greatest changes in magnitude are predicted by Web-
DO$_3$SE (5.1 mmol $O_3$ m$^{-2}$; 23.7%), followed by CMAQ_P (3.1 mmol $O_3$ m$^{-2}$; 16.7%), ZHANG (1.9 mmol m$^{-2}$;
14.1 %) and TEMIR (1.7 mmol $O_3$ m$^{-2}$; 9.6 %). In contrast, not all models are sensitive to changes of soil water
content (SWC). The greatest response is seen in CMAQ_P (-6.3 and +1.4 mmol m$^{-2}$; -34.0% and +7.6%),
followed by Web-DO$_3$SE (-2.2 and -2.2 mmol m$^{-2}$; -10.2% and -10.2%), and TEMIR (-1.1 and +0.8 mmol $O_3$
m$^{-2}$; -5.9% and +4.3%), while ZHANG shows no difference in this regard because it is not sensitive to soil
moisture. The changes are largest at crops (1.5 mmol $O_3$ m$^{-2}$; 13.4%), while grass and forest show similar
responses (2.8 and 1.7 mmol $O_3$ m$^{-2}$; 10.5 and 10.1 %, respectively). That is in line with De Marco et al. (2020)
who show that $POD_y$ responses to soil water changes increase with higher Y threshold (here crops). The models
do not appear to be sensitive to LAI 14d shifts, with the only exception of Web-DO$_3$SE, which simulates a lower
$POD_y$ for both early and late LAI shifts (-2.6 mmol $O_3$ m$^{-2}$ on average, across all land covers). LAI is used as a
proxy for growing seasons in most models whereasWeb-DO$_3$SE considers growing seasons directly.



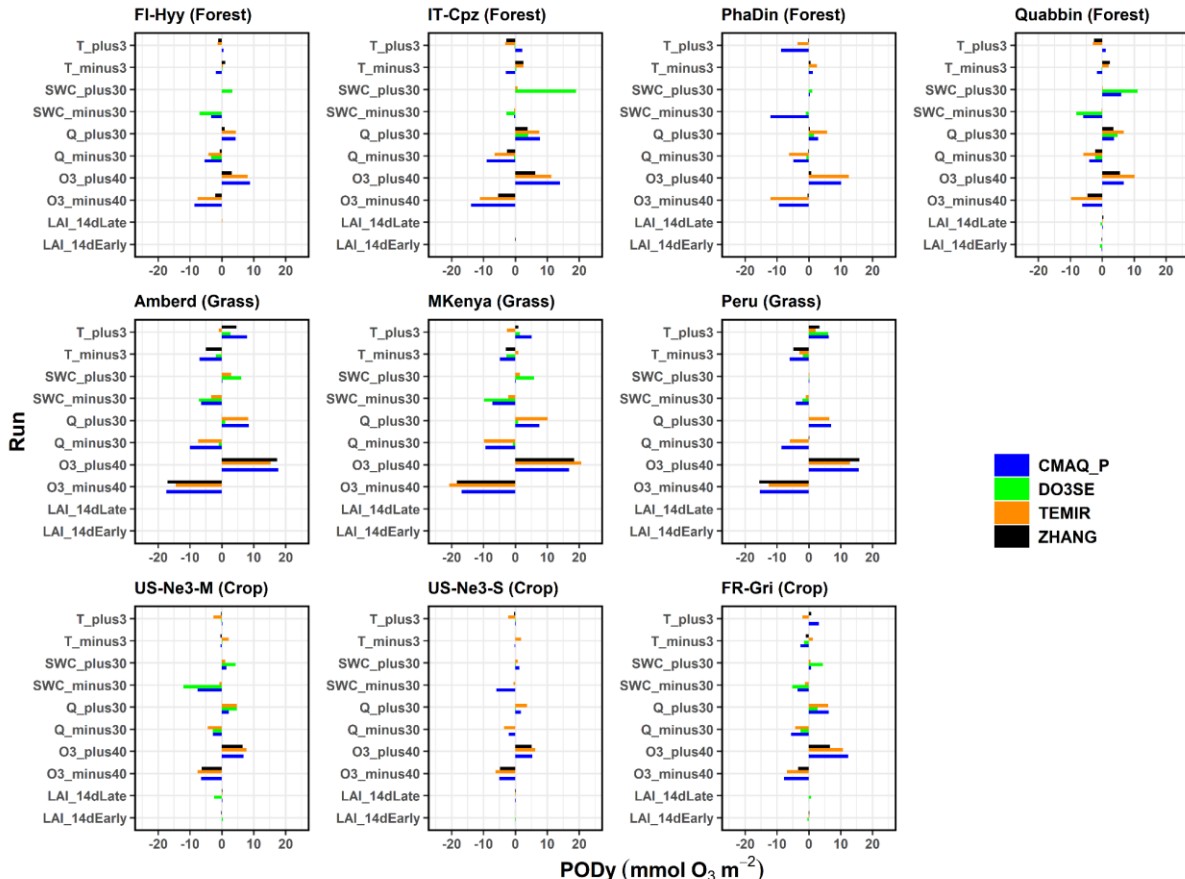


**Fig 8: Meteorology sensitivity assessment: Absolute change of POD$_y$ values with respect to Base run POD$_y$ due to 10 or 20 % variation of the temperature (T), soil water content (SWC), absolute humidity (Q), O$_3$ and LAI/growing season.**


A 20% change of g$_{max}$/Vc$_{max}$ leads to corresponding changes in POD$_y$ values. An increase or decrease of the
parameter leads to very similar changes (in +-) (Fig. 9, corresponding data in Table S12 – S14). The response
appears to be generally uniform across sites. On average, the results show +28.9 ± 22.4 % POD$_y$ change for the
20 % increase of g$_{max}$/Vc$_{max}$, and -27.4 ± 13.1 % for the 20 % decrease with the largest absolute changes at
grassland (up to 8 mmol m$^{-2}$ , ZHANG). At forests and crops, changes up to 5 and 3 mmol m$^{-2}$ occur,





respectively. Among all sites, noticeably higher (the highest) relative changes were estimated at FR-Gri which
thus constituted the only relevant source of variability. This change is significantly different to the change at US-
Ne3 (20-30 %) which reflects the contrasting low $O_3$ level at US-Ne3 compared to the highly polluted FR-Gri
site. Also, the ZHANG model predicts the highest changes at crops while CMAQ_P seems insensitive. The
ZHANG (and TEMIR) model appears to be the most sensitive model to the changes at most sites due to the
strong dependency on the $g_{max}$/$Vc_{max}$ parameter (see analysis above). The only climate trend of the response is
seen by the ZHANG model which shows an average 65 % increase/decrease in wet forests (PhaDin, FI-Hyy) and
only a 40 % change in dry places. Sites with very low estimates (PhaDin in ZHANG, Peru in Web-DO$_3$SE) were
excluded from this sensitivity study.

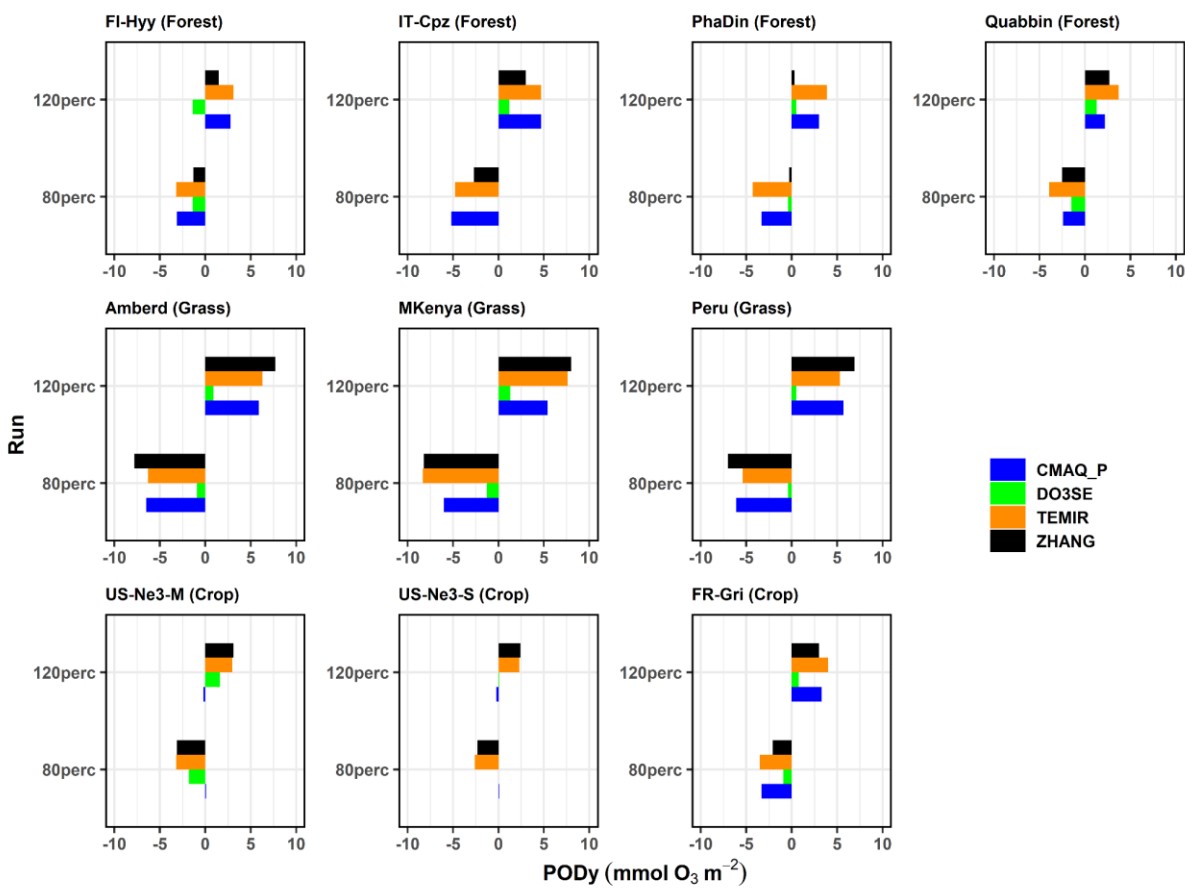






**Fig 9: Land cover parameterisation sensitivity assessment: Absolute change of POD$_y$ values with respect to the base run POD$_y$ values due to 20 % variation of Gsmax or V$_{Cmax}$.**

To indicate the likely damage, and range of damage that our modelled values of POD$_y$ predict, we have used POD$_y$ flux-response relationships available in the literature that most closely represent the vegetation type and climatic location of each study site (Table 5). To estimate O$_3$ damage to forests we use recently derived flux-response relationships that relate POD$_1$ values to gross annual increment (Karlsson et al., sub) and hence indicate the annual change in growth rate caused by O$_3$. The mean model ensemble estimates a percentage reduction in gross annual increment of around 5% for FI-Hyy and Pha Din, 6% for IT-Cpz and 14% for Quabbin. However, the range in estimates across models is not insignificant and most extreme at the Quabbin site with a minimum of 11% and a maximum of 21% around the mean 13% value; this is due to broadleaf deciduous species being more sensitive to O$_3$ dose than needleleaf species and hence more sensitive to a range of POD$_y$ model simulations (Bueker et al., 2015). It should also be emphasised that the Pha Din site uses a European-derived flux-response relationship for an Asian forest site.



| Site | Species | y | Flux-response relationship | Response metric & species | PODy | | | % Response | | | Location of PODy relationship | Reference |
| | | | | | min | median | max | min | median | max | | |
|---|---|---|---|---|---|---|---|---|---|---|---|---|
| FI-Hyy | Scots pine | 1 | y = -0.0057x + 1.0015 | Gross Annual Increment (GAI) % for Norway spruce/Scots pine | 2.3 | 10.2 | 15.1 | 1.2 | 5.6 | 8.5 | Europe | Karlsson et al., sub (to TOARII community special issue) |
| Quabbin | Birch/Beech (Broadleaf deciduous) | 1 | y = -0.0093x + 0.9461 | Gross Annual Increment (GAI) % for Birch/Beech | 6.5 | 9.1 | 16.8 | 11.4 | 13.9 | 21.0 | Europe | Karlsson et al., sub (to TOARII community special issue) |
| PhaDin | Norway spruce (Evergreen needleleaf) | 1 | y = -0.0057x + 1.0015 | Gross Annual Increment (GAI) % for Norway spruce/Scots pine | 0.4 | 8.1 | 20.3 | 0.0 | 4.5 | 11.4 | Europe | Karlsson et al., sub (to TOARII community special issue) |
| IT-Cpz | Holm oak | 1 | y = -0.0047x + 1.001 | Gross Annual Increment (GAI) % for Aleppo pine/Holm Oak | 2.8 | 13.3 | 25.8 | 1.2 | 6.2 | 12.0 | Europe | Karlsson et al., sub (to TOARII community special issue) |
| FR-Gri | winter wheat | 6 | y = -0.0385x + 1.003 | % grain yield loss for wheat | 3.6 | 6.8 | 9.3 | 13.6 | 25.9 | 35.5 | Europe | UNECE LRTAP Mapping Manual (2017) |
| US-Ne3 (Maize) | Maize | 6 | y=0.0426x+1 | % grain yield loss for wheat | 10.5 | 12.4 | 13.6 | | | | | Peng, J., Shang, B., Xu, Y., Feng, Z., Pleijel, H. and Calatayud, V., 2019. Ozone exposure-and flux-yield response relationships for maize. Environmental pollution, 252, pp.1-7. |
| US-Ne3 (Soybean) | Soybean | 6 | y = -0.033x + 1.01 | % relative seed yield loss per soybean plant | 0.0 | 8.3 | 11.0 | 0.0 | 26.4 | 35.3 | China | Zhang, W., Feng, Z., Wang, X., Liu, X., Hu, E. (2017) Quantification of ozone exposure-and stomatal uptake-yield response relationships for soybean in Northeast China. Sci of the Total |





| | | | | | | | | | | | | | | |
|---|---|---|---|---|---|---|---|---|---|---|---|---|---|---|
| | | | | | | | | | | | | Env., (710-720) | 599-600 | |
| Amber d | Grassland | 1 | y = -0.0062x + 0.947 | % total biomass loss for temperate perennial grassland | 7.9 | 29.4 | 34.1 | 10.2 | 23.5 | 26.4 | Europe | UNECE Mapping (2017) | LRTAP Manual |
| MKeny a | Grassland | 1 | y = -0.0062x + 0.947 | % total biomass loss for temperate perennial grassland | 10.9 | 31.0 | 37.4 | 12.1 | 24.5 | 28.5 | Europe | UNECE Mapping (2017) | LRTAP Manual |
| Peru | Grassland | 1 | y = -0.0062x + 0.947 | % total biomass loss for temperate perennial grassland | 2.6 | 22.1 | 26.6 | 6.9 | 19.0 | 21.8 | Europe | UNECE Mapping (2017) | LRTAP Manual |

**Table 5. Estimates of O₃ damage (for specific response metrics) derived from using the ensemble mean modelled PODy**
**values (and minimum and maximum values) with appropriate flux-response relationships based on land cover type. The**
**climatic location within which the flux-response relationships are derived are stated to show the relevance of their use in**
**estimating damage. Shaded cells denote flux-response relationships that are derived outside of the broad climate region**
**to which they are applied in this study and hence whose damage estimates should be treated with caution.**




For crops, flux response relationships are available for wheat, maize and soybeans (UNECE LRTAP, 2917, Peng et al., 2019
and Zhang et al., 2017). These relationships are derived from Europe (wheat) and China (maize and soybean). For wheat, we
see a large range in percentage yield loss with a mean model ensemble of 26 % but a maximum yield loss of 35 %. This is
driven by high $POD_6$ values derived from CMAQ_P and TEMIR. For maize at US-Ne3 the results are very consistent with
relative grain yield loss estimates ranging from 1.4 to 1.6 %. For soybeans at US-Ne3, the results are less consistent than
maize with a minimum and maximum of 0 and 35 % yield around a mean of 26 %. It is important to note that a Chinese-
derived flux-response relationship is used to estimate O₃ damage on both US-grown crops.



Finally, for grasslands, we estimate total biomass losses of 19, 24 and 23% from the ensemble model mean for Peru, Mt
Kenya and Amberd respectively. The range in model values is relatively small for Amberd and Mt Kenya. A low minimum
value of 6 % total biomass loss is estimated for Peru due to the Web-DO$_3$SE model having a very low POD$_y$ at this location
due to a likely oversensitive limitation to O$_3$ uptake caused by low temperatures.

## 566  4. Discussion and Conclusion

Here we have compared six deposition schemes commonly used in atmospheric chemistry transport models. We have
focussed on the stomatal component of deposition since this is acknowledged to have a substantial influence on damage to
vegetation, and ultimately the ability of these six models to estimate the POD$_y$ metric designed to indicate the level of O$_3$
damage to forest, crops and grasslands. The models estimate POD$_y$ values of 28, 15 and 9 mmol O$_3$ m$^{-2}$ for grassland, forests
and crops, respectively. The multi-model mean estimates are generally in the expected range which suggests that the
stomatal flux output of these models could be used for O$_3$ impact assessments. We also explored the differences in POD$_y$ by
geographical location. When comparing one vegetation type, we find multiple drivers including O$_3$ concentration. The
different model types are not the driving force, instead, the models can predict similar results.
There are three key reasons for differences in dry deposition model estimates i. model construct and the inclusion/exclusion
of important factors that determine G$_{st}$ and G$_{sun}$; ii. model parameterisation which may characterise the land cover types and
iii. differing model sensitivity to climate variables (seasonal, location effects) in estimates of stomatal deposition. The model
comparison of stomatal conductance and stomatal dry deposition for ozone helps us to understand the differences between
models. We found that models simulate generally reasonable stomatal deposition of 0.5 -0.8 cm s$^{-1}$ in summer whereas the
different model types often agree very well with each other. The stomatal conductance estimates among the models agree
with correlation coefficients of 0.75, 0.80 and 0.85 for forests, crops and grasslands. The model differences, identified during
this analysis, can be explained by the model's dependence on the meteorological conditions at sites. Indeed, both model
structure (e.g. Raghav, Kumar and Liu 2023) and parameters (Fares et al., 2013) can affect the accuracy of stomatal
conductance models. However, studies have shown that when properly calibrated against field observations, structurally
different stomatal models can produce similar stomatal conductance (Fares et al., 2013, Mäkela et al., 2019). Calibrating the
key parameters of stomatal conductance models (e.g. g$_{max}$/Vc$_{max}$) is a crucial next step to improve the accuracy of stomatal
conductance and POD$_y$ estimates, as our sensitivity tests show direct, and possible non-linear relationship between POD$_y$ and
g$_{max}$/Vc$_{max}$ (e.g. at FR-Gri). This is possible with the recent availability of standardised global eddy flux (FLUXNET,
Pastorello et al., 2020) and sap flow (SAPFLUXNET, Poyatos et al., 2020) data.



To estimate $POD_y$ for a representative leaf of the upper canopy, the sunlit leaf must be distinguished from the total leaf.
Since the effects-based community recognised that sunlit leaves contribute most to carbon assimilation throughout the
growing season or $O_3$-sensitive period (e.g. in wheat, this is considered to be the time from anthesis to maturity) and hence it
will better represent damaging $O_3$ uptake. All flux response relationships for $POD_y$ are developed for such a representative
leaf. This is an important distinction since previous model comparison studies (e.g.Clifton et al., 2023) have tended to focus
on whole canopy dynamics. These are important to estimate accurately, but to estimate $POD_y$ requires additional canopy
level processes, which need i. $O_3$ concentration at the top of the canopy, ii. wind speed at the top of the canopy and iii. $G_{sun}$
of a representative leaf at the top of the canopy.
Our models estimate 30-50 % of stomatal $O_3$ deposition at sunlit leaves. Thereby, the model estimates of the total stomatal
flux are more widespread (during one season) than the estimates of the sunlit only which suggests an important role of the
model's partitioning in two big leaves. When calculating $POD_y$ model means estimates generally agree with the literature but
most discrepancies between model estimates of $POD_y$ ultimately come down to the differences in simulations of stomatal
conductance. The sensitivity analysis of $POD_y$ yields ozone as the most important input variable, to whose changes all
models respond similarly. Considering all models and sites together, $POD_y$ were affected most by the $O_3$ concentration (+-
60-80 % site-dependent, i.e., higher $O_3$ conc leads to higher $POD_y$), followed by humidity (30-50 % site-dependent impact).
Soil moisture impacts were also significant for the CMAQ_P and Web-DO$_3$SE model (up to +-68 % and 22 % change). The
sensitivity to temperature changes varies strongly among the model and its parametrization. As the plant canopy acts as a
persistent sink of  $O_3$, there is a significant vertical gradient of  $O_3$ within the atmospheric surface layer. For example, Travis
et al. (2019) show that the midday $O_3$ concentration at 65 m above ground (mid-point of a first vertical layer of GEOS-Chem
v9-02) is 3 ppb higher than the $O_3$ concentration at 10 m above ground (inferred by Monin-Obukhov Similarity Theory,
MOST) over the Southeastern United States. A mismatch between $O_3$ measurement height and canopy height can lead to
inaccurate $POD_y$ calculation (Gerosa et al, 2017). As we show that the errors in $O_3$ concentrations propagate non-linearly to
$POD_y$ (i.e. 40% changes in $O_3$ leads to 53 - 68 % changes in $POD_y$), such a mismatch should be carefully avoided by
applying atmospheric surface layer theories (e.g. MOST) to estimate the vertical profile of  $O_3$, and therefore the canopy-top
$O_3$ concentration, if direct measurement or model output of $O_3$ at canopy top is not available.

Finally, we use flux-response relationships for temperate deciduous (Beech/birch), temperate needleleaf (Norway spruce
(*Picea abies*)), crops (wheat (*Triticum aestivum)*, maize (*Zea mays*) and soybeans (*Glycine max*)) and grassland (*Lolium*
*perenne*) to give a suggest the potential likely variation of damage estimates by land cover type and climatic region. These
relationships have predominantly been developed for European and Asia forest and crop species. Therefore, they should be
applied to other climate regions with caution although recent evidence suggests that tropical forest species may have similar



sensitivity to $O_3$ as European species (Cheeseman et al. 2024). Although there is rather large variability in $POD_y$ values
estimated by the model, the median values are relatively robust. Unfortunately, there is only statistical or modelled evidence
of actual $O_3$ damage, and only at a few of the sites investigated. Modelled evidence uses stomatal ozone flux models similar
to those used in this study, but which have been parameterised for local site conditions (Stella et al., 2013 for FR-Gri wheat).
Simulations with a terrestrial biosphere model suggested an average long-term $O_3$ inhibition of 10.4% for the period 1992–
2011 at the Harvard site (Yue et al 2016); this compares to our model ensemble estimate of 14% GAI biomass loss for
Quabbin. A significant but small NEP reduction was found during Spring in the Italian Castelporziano forest site (up to -
1.37 %) but not at the FI-Hyy or FR-Gri sites (Savi et al., 2020). Our modelling estimated substantially lower $POD_y$ values
and associated damage at Hyy and IT-Cpz than Quabbin though we would expect to see a more substantial $O_3$ effect than
that demonstrated by the NEP statistical modelling (i.e. 5 and 6% GAI biomass loss at FI-Hyy and IT-Cpz respectively).
Similar simulations with a different terrestrial biosphere model found only moderate $O_3$ damage effects (GPP reductions of
4–6 %; Yue & Unger, 2014). This result is driven by low ambient ozone concentrations but also by the choice of a C4
photosynthetic mechanism to estimate stomatal conductance which gives relatively high-water use efficiency). These
simulations also suggested that the US-Ne3 experienced a higher ozone effect on GPP than Harvard which is consistent with
our modeling for soybeans (but not maize, generally considered an $O_3$ tolerant crop species; Mills et al 2011). According to
the $POD_6$ estimates made using a SURFATM model, parameterised for Grignon wheat, $POD_6$ values of 1.094 mmol $O_3$ m$^{-2}$
were estimated from 1 April to 1 July 2009 which compared with our range of 3.6 to 9.3; the locally parameterised values
gave estimated crop yield losses of 4.2%, compared to our median model ensemble estimates of 25% for the winter wheat.
This is most likely due to the lower $g_{max}$ value used in the local parameterisation (296 mmol $O_3$ m$^{-2}$ s$^{-1}$). However, no
recording of actual damage is given at the FR-Gri site, so it is not possible to tell which of these simulated damage estimates
is closer to reality.
The experiments performed here with varying climate and vegetation input data also find a similar sensitivity of PODy to $O_3$.
It is helpful to have a range of models and model constructs in deposition schemes especially where these have been
developed for particular land cover types. When used in damage estimates it is important to ensure that key stressors are
included which may be important for that respective geographical region (such as soil and vapour pressure deficit).
Recognising that several deposition schemes would be able to reliably predict PODy for different climates and cover types
once they have been parameterised appropriately will extend the usefulness of flux-response relationships.
All in all, we have demonstrated, through this paper, the widespread applicability and consensus among various numerical
stomatal flux methods and identified the key model constructs and parameterisations that cause differences in ozone
deposition and PODy estimates. Our results and findings present exciting opportunities, enabling us to extend the application
beyond specific sites and growing seasons, to conduct comprehensive global stomatal flux studies over long periods.



Integrating the TOAR database with the Web-DO₃SE model enables automatic models runs for ozone-vegetation impact
assessment at a large range of sites using the TOAR database.


**Author contributions**
T.E.: site selection, TOAR data extraction, data preparation, model support, modelling Web-DO₃SE, writing, coordination.
A.M.: modelling (ZHANG, MESSy, NOAH-GEM, TEMIR model), statistics, plots and analysis. L.E.: concept, writing.
H.M.: writing, reviewing. L.Z.: concept and writing. L.R: modelling with CMAQ, FLUXNET data preparation. C.B.:
debugging and test simulations of Web-DO3SE. A.W.: site selection, preparation of FLUXNET and sensitivity data. G.K.:
site selection, TOAR data extraction. G.G.: site analysis. M.H.: plots and reviewing. P.G.: PODy analysis.

**Competing interests**
The authors have no competing interests.

**Acknowledgements**
We acknowledge the TOAR team supports the data extraction. The authors acknowledge the access to the meteorological
data on the Jülich MeteoCloud provided by Jülich Supercomputing Centre (Krause et al., 2018). We thank the responsible
people of the selected measurement sites for their support in obtaining site information. We greatly appreciate helpful
discussions in the earlier stages of the project from the following people: Owen Cooper, Zhaozhong Feng, Laurens
Ganzeveld, Meiyun Lin, Martin Schultz, Eran Tas, and Oliver Wild.

**Code availability**
The Web-DO₃SE source code is freely available at https://toar-data.fz-juelich.de/ under the CC-BY 4.0 license
(https://creativecommons.org/licenses/by/4.0/). The further model code can be obtained upon request.

**Data availability**
The TOAR data is freely available at https://toar-data.fz-juelich.de/ under the CC-BY 4.0 license
(https://creativecommons.org/licenses/by/4.0/). The ERA5 data used can be downloaded from the MeteoCloud server
(https://datapub.fz-juelich.de/slcs/meteocloud/index.html). FLUXNET 2015 dataset is publicly available at
https://fluxnet.org/data/fluxnet2015-dataset/. Stomatal conductance estimates, and the related FLUXNET 2015 data from
SynFlux version 2 can be obtained by contacting Christopher Holmes (cdholmes@fsu.edu).



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






**Appendix**

Table A1: Abbreviations

| Symbol | Long name |
|---|---|
| rsmin | Minimum stomatal resistance in [s m$^{-1}$] |
| gsmax | Maximum stomatal conductance in [m s$^{-1}$] |
| RH | Relative humidity in [%] |
| LAI | Leaf area index in [m$^2$ m$^{-2}$] |
| sd, sn | snow depth in [m] and snow cover |
| ssrd, strd | solar and thermal flux at surface in [W m$^{-2}$] |
| sw | Soil wetness [m] |
| al_vis: | albedo (visible) |
| cwv | canopy water content in [kg m$^{-2}$] |
| SWC | Soil water content |
| SM | Soil moisture [m$^3$m$^{-3}$] |
| wdir | geo wind direction [°] |
| wspeed | Wind speed in [ms$^{-1}$] |
| cv | Vegetation fraction [m$^2$m$^{-2}$] |
| P | Precipitation in [mm] |
| P_rate | Precipitation rate in [mm h$^{-1}$], [kg m$^{-2}$ s$^{-1}$], [m s$^{-1}$] |



| Tair, Tsoil, T2m | Air, soil, 2m temperature in [K] |
|---|---|
| VPD | Vapour pressure deficit [kPa] |
| Pa | Air pressure [hPa] |
| Rn, Gr | Net and global radiation [W m$^{-2}$] |
| u* | Friction velocity [m s$^{-1}$] |
| O3, CO2 | $O_3$ and $CO_2$ concentration in [ppb] und [ppt] |
| h_dis, z0 | Displacement height [m], roughness length [m] |
| CF | Cloud fraction |
| LUC | Land usage category |

