# Peer review of "Can atmospheric chemistry deposition schemes reliably"

_EGUsphere, 2025_

## Author Comment (AC1)

**Point-by-point reply**
**Reviewer 1**

Dear Authors,
The manuscript "Can atmospheric chemistry deposition schemes reliably simulate stomatal ozone flux across global land covers and climates?" presents a thorough and well-structured analysis of ozone deposition and its implications for vegetation. The study is not only scientifically rigorous, but also highly relevant to understanding how different empirical and semi-empirical modeling approaches influence the assessment of air pollution effects. The analysis is very interesting and well-structured and the combination with a clear exposition of key findings, such as variability in stomatal and non-stomatal deposition and the impact of meteorological factors on $O_3$ uptake, makes this paper a valuable contribution to the field.

The authors have done an excellent job of detailing the methodology, providing transparency in comparing the models. Furthermore, the manuscript effectively highlights the strengths and limitations of different modeling approaches; for example how the models incorporate detailed land cover parameters while others rely only on more generalized assumptions, influencing deposition estimates.This nuanced perspective enhances its scientific relevance. The discussion is well structured and the results are presented in a way that facilitates interpretation, making this study a significant reference for future research on air pollution and its environmental impacts. So I suggest only a minor revision just to make the whole paper clearer and more fluent.

We gratefully acknowledge the reviewer's positive feedback and have addressed individual comments carefully. You will find our answers highlighted in blue and changes in red below.

**Abstract**
The abstract is dense and difficult to read, probably due to an excessive amount of condensed information. To improve clarity, it would be useful to simplify the sentences, reducing their length and complexity, and to reorganize the content in a more linear way. A clearer structure, which clearly distinguishes the context, objectives, methods, main results, and implications of the study, would help the reader to grasp the essential message more easily. Furthermore, eliminating secondary details would make the text more fluid and immediate, without compromising the completeness of the information.

We strongly agree with the reviewer and follow the suggestions to improve the readability. We change the abstract to the following:
„Over the past few decades, ozone risk assessments for vegetation have evolved two methods based on stomatal $O_3$ flux. However, substantial uncertainties remain in accurately simulating these fluxes. Here, we investigate stomatal $O_3$ fluxes across various land cover types worldwide simulated by six established deposition models. Hourly $O_3$ concentration and meteorological data at nine sites were extracted from the Tropospheric Ozone Assessment Report database, a comprehensive global collection of measurements, for the model simulations. The models estimated reasonable $O_3$ deposition (0.5 - 0.8 cm $s^{-1}$ in summer) which is mostly in agreement with the literature. Simulations of canopy conductance showed differences of models that varied by land cover type with correlation coefficients of 0.75, 0.80 and 0.85 for forests, crops and grasslands among the models. Differences between models were primarily influenced by soil moisture and vapor pressure deficit (VPD), depending on each model's specific structure. Across models, the range of $O_3$ damage ($POD_y$) simulations at each site was most consistent for crops (6 to 11 mmol $O_3$ $m^{-2}$), followed by forests (3 to 19.5 mmol $O_3$ $m^{-2}$) and grasslands (7 to 33 mmol $O_3$ $m^{-2}$). The median estimate across models aligns well with the literature at the sites most vulnerable to $O_3$ damage. Overall, this study

represents a critical first step in developing and evaluating tools for broad-scale assessment of $O_3$ impacts on vegetation within the framework of TOAR phase II."

**1.Introduction**

The introduction could be strengthened by discussing the broader implications of the differences between the models for real-world applications, particularly in the context of ecosystem management. And to give the reader a general understanding of whether using one model is preferable to another under certain conditions.

In lines 71- 81, we describe the commonly used models: 'The common Jarvis-type models (e.g. Emberson et al., 2000; Ganzeveld et al., 1995; Zhang et al. 2003), widely applied due to their simplicity and computational efficiency, correct a prescribed maximum stomatal conductance with the multiplication of different environmental factors (e.g., temperature, light, soil water and atmospheric moisture). The $A_{net}$-$g_s$ models couple $g_s$ to plant photosynthesis by calculating the net assimilation of $CO_2$ and estimating gs based on the resulting supply and demand of $CO_2$ (Farquhar et al., 1980; Goudriaan et al., 1985; Ball et al., 1987). $A_{net}$-$g_s$ models involve multiple non-linear dependencies on soil water, humidity and temperature, among other factors defined by measurement constraints (Ball 1987; Leuning et al., 1997). Heterogeneity of stomatal deposition estimates over different land cover types is anticipated, but model uncertainty depends on the representation of the deposition mechanisms, model parameterisation and meteorological inputs (Hardacre et al., 2015; Clifton et al., 2020b; Huang et al., 2022; Khan et al., 2024).

According to the reviewer's comment, we add here the following: 'Broadly speaking, the pros and cons of these two modelling approaches will tend to depend on the aims of the risk assessment study, the extent of knowledge of the ecosystem being investigated and prevailing bio-climatic conditions. Jarvis-type models are arguably more suitable for studies where less is known about the eco-physiology of the ecosystem since they do not require simulation of net photosynthesis which in itself is inherently difficult to model accurately. However, these models still need to be calibrated for the particular bio-climate of study to ensure temperature and VPD functions are suitable for the prevailing conditions. By contrast, Anet-gsto models may be more useful for studies where the physiological response to environmental conditions of the ecosystems is reasonably well understood as they can provide insight into not only pollutant deposition, but also how other environmental conditions in addition to pollution may limit plant growth and productivity more generally. '

**2. Methodology**

The methods are mostly well described, but clarity on the consistency of input data across models (e.g., meteorological forcing, land cover) would improve transparency.

We add at the beginning (l. 99) the following:
"The web version of the DO3SE model is coupled to the TOAR database, i.e. the required input data (Table 3) is automatically provided by the database at the respective modelling sites."

Line 116, we expanded the description as follows:
"To conduct offline simulations with models in addition to Web-DO3SE, the input data were extracted beforehand and proven for identicality. The additionally required data (Table 3) were extracted from the TOAR database and the MeteoCloud server (https://datapub.fz-juelich.de/slcs/meteocloud/index.html) at Forschungszentrum Jülich."

The role of non-stomatal deposition (cuticular, soil absorption) could be more explicitly detailed.

The manuscript is focused on stomatal deposition, because it is aimed to assess the vegetation impact. To be able to classify the contribution of the stomatal pathway with the total deposition at the individual sites we use the concept of effective conductances (including cuticular and soil deposition, SI equations 3-5). For the detailed calculations of cuticular deposition, we refer to the individual model descriptions. Most models only consider a general representation of soil deposition (no explicit calculation).

While the methodology is scientifically sound and well documented, the description of data selection criteria could be more explicit, especially regarding how site inclusion decisions affect the analysis. While the study acknowledges potential errors in the TOAR-II database, adding a small explanation would enrich the manuscript, especially regarding implications for the results.

The criteria (ensuring the selection of different plant functional types and land cover types which also had O3 and meteorological data and ideally gst measurements) limited the site selection seriously. In the analysis, we interpret the results always with regard to the site-specific conditions. To further clarify that the site selection influence the interpretation of the results, we add to line 580 in the discussion:
[...with correlation coefficients of 0.75, 0.80 and 0.85 for forests, crops and grasslands]. "Thereby, the 9 sites selected for this study also reflect different climate conditions; however the selection of sites that provide such broad representations also means that the analysis and the results cannot be generalized. The global coverage, diverse land types and varying meteorological conditions of the 9 sites resulted in widespread model responses to soil moisture (Fig. 8), while appearing to be insensitive to changes of LAI (Fig. 9). The former underscored the idiosyncratic features and hence potential limitations of individual models, whereas the latter gave us confidence in model capabilities despite the different constructs and parameterizations of the models. "

The potential errors in the TOAR database are small and have been shown to have a minor impact (Schultz et al., 2017, Tarasick et al., 2018), which is described in lines 106-108. Therefore, we expect no significant impact on the model results. In line 611 to 614 (discussion section), we elaborated on uncertainties in estimating O3 deposition in the original manuscript, and now add more information according to the reviewer's comment: 'For example, Travis et al. (2019) show that the midday $O_3$ concentration at 65 m above ground (mid-point of a first vertical layer of GEOS-Chem v9-02) is 3 ppb higher than the $O_3$ concentration at 10 m above ground (inferred by Monin-Obukhov Similarity Theory, MOST) over the Southeastern United States. A mismatch between $O_3$ measurement height and canopy height can lead to inaccurate $POD_y$ calculation (Gerosa et al, 2017). An O3 bias of 2 ppb as estimated by e.g. Tarasick et al. (2018) would lead to a change of 6-7 % in POD1 (Gerosa et al., 2017). Similarly, we show [...]'

**2.3 Stomatal deposition models and their key inputs**
L 158-161: The explanation of the models is done in quite a bit of detail, but the reason why these models were chosen over others is not so clear, it could be explained better.

L 169: I don't understand why some model descriptions are more detailed than others. For example, the MESSy model approach is mentioned briefly compared to others. Standardizing the level of detail for each model would improve readability.

In the model description, we summarize the key features and the specialities of each model. To emphasize the model differences we modify the description to:
 The key model features are described below.

(1) The empirical/Jarvis-type models use a predefined stomatal conductance modified with different environmental stressors for radiation (PAR), air temperature (T), vapour pressure deficit (VPD) and soil water (SM) - The ZHANG model (Zhang et al., 2002; 2003; 2006) and the Web-DO₃SE model (i.e., a version of DO₃SE that is directly coupled to the TOAR database, Emberson et al. 2000) account for sunny and shaded leaves (two-big leaf), the Web-DO₃SE model additionally depends on the vegetation phenology, the CMAQ_J model (Pleim and Ran, 2011) and the MESSy model (Ganzeveld et al., 1995; Kerkweg et al., 2006) account for one-big leaf  CMAQ_J uses relative humidity (RH) instead of VPD. MESSy calculates the initial stomatal conductance based on the PAR and several empirical parameters

(2) Semi-empirical/Ball-Berry - The CMAQ_P model  (Ran et al,. 2017) and the TEMIR model (Collatz et al., 1991; Farquhar et al., 1980) calculate the stomatal conductance at sunlit and shaded leaves for C3 and C4 plants depending on net CO2 assimilation rate, CO2 partial pressure, atmospheric pressure (Pa) and water vapor pressure for each leaf. The  NOAH-GEM model is different, calculating the stomatal conductance at one big leaf using RH instead of VPD  (Wu et al., 2011; Niyogi et al., 2009).

L 178-181: The resistance pattern is not immediately clear; I suggest a modification to improve its clarity.

We suggest here below an improved scheme:

[Figure]

L 275-282: It is mentioned that factors such as wind speed and solar radiation influence O₃ deposition, but the explanation is not very clear and fluid. It might be useful to introduce a transition sentence to clearly highlight how each factor impacts the different components of deposition.

Yes we agree, and change line 274/275 to: 'At the Amberd and Peru sites, $G_{cut}$ and $G_{ground}$ are low since low wind speed reduces downward mixing of ozone to the surface (atmospheric resistance).'
Line 278, we modify to: 'In contrast, at the Mt Kenya site, $G_{st}$ exceeds $G_{cut}$ and $G_{ground}$, since the strong solar radiation (annual mean is 246 W m$^{-2}$, Table S2) at this site favours stomatal opening . Besides that, LAI is a very important governing factor for $G_{st}$ . Therefore, it can be inferred that the O₃ deposition pathway depends on not only the land cover type but also meteorological drivers. The relative contributions of each deposition pathway depends on the interplay between these key factors at a particular site.'

**3.2 Vegetation impact and variation with key input data**
In this section, the terms PODy, POD1 and POD6 alternate without a clear transition between them, which could generate confusion in the reader. It would be useful to introduce a more structured analysis of the differences between these indicators. For example, specifying what their implications are in the different ecosystems analyzed. Furthermore, a more fluid connection between the various metrics would help to make the analysis clearer and easier to understand.

We thank the reviewer for this valuable suggestion and introduce a description of the different metrics in line 435: 'The critical threshold for ozone damage y differs for the three land cover types. For forests and grass the y value is 1 nmol $O_3$ m$^{-2}$ s$^{-1}$ (POD1), O3 damage to crops is assumed to occur only when the y threshold exceeds 6 nmol $O_3$ m$^{-2}$ s$^{-1}$ (POD6).'
From there onwards, we change 'PODy' to the respective term POD1 or POD6.

**4. Discussion and Conclusion**
In this section the obtained results are clearly reported, only some passages between the topics (for example, from the discussion of the models to the division between leaves exposed to the sun and shaded) could be made more fluid with connecting sentences.

We change line 329/330 to: 'Here, it is important to understand the model distinction between shaded and sunlit leaf ($G_{sun}$ , Fig 4)'.

Furthermore, more emphasis could be given on practical implications: for example, the final section could be slightly expanded to underline the impact of the results.

We extend line 645 ff. as follows: 'Overall, this study has demonstrated the widespread applicability and consensus among various numerical stomatal flux methods. Both semi-mechanistic as well as empirical models can generally represent observed ozone fluxes among different land cover types and climates. We identified the key model constructs and parameterisations that cause differences in ozone deposition and PODy estimates. However, none of the models clearly shows a superior overall performance. Instead, all models can be effectively applied, each with its own strengths and weaknesses. Our findings present exciting opportunities to extend applications beyond specific sites and growing seasons, enabling comprehensive global stomatal flux studies over longer periods. '

---

## Author Comment (AC2)

**Reply to Reviewer 2**

**General assessment:**

This well-written and timely study evaluates the capacity of six widely used ozone deposition models to simulate stomatal $O_3$/fluxes across various global land cover types. The manuscript contributes to the Tropospheric Ozone Assessment Report (TOAR-II) community effort by assessing model behaviour under standardised conditions. It also explores both inter-model variability and sensitivity to key drivers. The study is particularly relevant for improving global ozone risk assessments and advancing vegetation impact modelling. The integration of FLUXNET and SynFlux observational constraints is commendable, and the structured multi-experiment framework is a strong point of the manuscript. That said, several aspects require clarification, particularly around the interpretation of model differences, treatment of uncertainties, and consistency in terminology and figures.

We gratefully acknowledge the reviewer's positive feedback and have addressed individual comments carefully. You find our answers highlighted in blue and changes in red below.

**Major comments:**

**Clarity on Model-Observation Agreement:**

The evaluation of modelled Gst against SynFlux-derived values is informative, but the conclusions could be more precise. It's difficult to assess which model(s) perform best consistently across sites. A summary table with performance metrics for each site and model would strengthen this section.

Consider providing a visual summary (e.g. radar plot or heatmap) comparing model agreement with observations across all evaluated metrics.

Thanks for this valuable suggestion. We prepared scatter plots of hourly modeled stomatal flux and the respective SynFlux stomatal flux (Fig. S3). Furthermore, we have heatmaps of spearman correlation coefficients (p<0.05) between the hourly canopy-level modelled Gst using all available data (including SynFlux) in the supplement information (Figures S4). We also created a table showing the normalized mean bias of each model against Synflux (Table S3).

At both sites, all models perform well with correlation between 0.65 - 0.85 whereas best values are reached by the TEMIR and CMAQ_P.

[Figure]

**Fig S4. Spearman correlation coefficients (p<0.05) between the hourly canopy-level modelled Gst using all available data (including SynFlux). Models were run from the FLUXNET input data.**

[Figure]

[Figure]

[Figure]

**Figure S3. Scatter plots of hourly modeled stomatal flux and the respective SynFlux stomatal flux. 'r' represents Spearman correlations between the hourly fluxes through the entire year (p < 0.05).**

**Table S5. Normalized mean bias (NMB; %) and Spearman correlation coefficient (r) of the model-predicted $F_{st}$ with respect to SynFlux $F_{st}$. Data for the entire year was used for the calculation.**

| Site | Model | NMB (%) | r |
|---|---|---|---|
| US-Ha1 (Forest) | CMAQ_J | -7 | 0.9 |
| | CMAQ_P | 54 | 0.81 |

| | | | |
|---|---|---|---|
| | MESSY | 180 | 0.78 |
| | NOAH | 10 | 0.79 |
| | TEMIR | 9 | 0.87 |
| | ZHANG | -12 | 0.81 |
| FI-Hyy (Forest) | CMAQ_J | 23 | 0.8 |
| | CMAQ_P | 84 | 0.73 |
| | MESSY | 222 | 0.63 |
| | NOAH | -14 | 0.75 |
| | TEMIR | 60 | 0.77 |
| | ZHANG | 4 | 0.71 |

**Treatment of Uncertainty:**

While uncertainty is addressed via sensitivity experiments and ensemble medians, explicit ranges or confidence intervals for key outputs (e.g., PODy estimates) across models would be useful.

Given this variability, how robust are the conclusions regarding PODy differences across land cover types?

Within the scope of the current study we were unable to perform a full sensitivity analysis on the PODy model outputs since this would have taken substantial time and computational power. Also, given the lack of knowledge over the probability distribution of key model parameters for the land cover types explored it would also have suggested a level of knowledge exceeding what we actually have. Therefore, we used the simplified sensitivity assessment to assess which were the key input variables and model parameters that would warrant further study. However, we feel that the PODy differences simulated between land cover types are relatively robust as we do have good evidence to show that key input data (e.g. growing seasons) and variables (e.g. gmax for multiplicative models and Vcmax for photosynthetic based models) are different between land cover groups which will drive broad differences in PODy values. Given the complexity of the deposition models (especially the photosynthesis-based ones), robust confidence intervals could only be computed by

Monte Carlo simulations, which is too computationally expensive since this requires running the models a few thousand times. Also, knowledge over the probability distributions of key model parameters would be required.

**PODy Thresholds and Flux-Response Relationships:**

The thresholds used for PODy calculation (e.g., 1 nmol m²s¹ for forests) are stated clearly, but are any species-specific or site-specific adaptations made? The text could benefit from a brief reflection on the limitations of using fixed thresholds across diverse vegetation.

The threshold y is the detoxification capacity, the chosen values are commonly used for crops, forests or grassland (Emberson 2020). In fact, we selected the 6 sites based on their land cover types and we applied the known threshold for the individual sites based on Emberson (2020). We add the following explanation in the respective text: "Studies (Emberson 2020 and references therein) have established thresholds for different land cover types which are used to provide y values for the selected sites with specific land cover types in this study. Some studies suggest that the y threshold for land cover types may vary by global region (e.g. a number of studies suggest higher y values of up to 12 nmol $O_3$ m$^2$ s$^{-1}$ is more appropriate for crops and forest tree species in Asia). In this study, which focuses on comparing across models, we maintain consistency and use common y threshold values for each landcover type. However, this is an aspect that would benefit from further study in the future since estimating PODy values with higher thresholds is more challenging for all types of model given the less frequent occurrences of such high $O_3$ doses. "

**Figures and Data Presentation:**

Figures 3–6 are central to the conclusions, but they are visually dense due to the number of sites and models. Consider moving some detailed seasonal panels to the Supplement and simplifying the main figures.

We moved the panels showing winter, spring and autumn in Fig. 3 and 4 to the supplement.

**Minor Comments:**

Ensure consistent use of chemical notation: Use subscript formatting (e.g., $O_3$, $CO_2$) where possible. Standardize units throughout the text and figures (e.g., "mmol $O_3$ m$^{-2}$" vs "mmol O3 m-2", "cm s$^{-1}$" vs "cm/s").

We harmonized the units according to Copernicus standards.

In multiple places, "sunlit" is referred to (e.g., Fst, sun, Gsun). Define these variables clearly in the main text, not just in figure captions or formulas.

The definitions can be found in line 332/333 ('This also helps interpret the modelled stomatal conductance of sunlit leaves ($G_{sun}$) shown in Fig 4.') and line 395-397 ('Figures 5 and 6 show the (SRAD>50 Wm$^{-2}$) stomatal $O_3$ flux ($F_{st}$) and stomatal, sunlit $O_3$ flux ($F_{st,sun}$) for different models per season at 9 sites representing forest (top), grass (middle), crops (bottom).'

Figures 3, 4: Increase font size in legends and axes for readability.

Done.

Figure 7–9: Consider sorting or grouping sites by land cover or latitude for more straightforward interpretation.

We understand that all figures are dense due to the multiple models used. To make interpretation easier, the same land cover type is displayed in one row.

Table 4: Clarify whether VCmax values refer to standardised temperature conditions (25°C). Also, state if values are per sunlit leaf area or total canopy.

Yes, VCmax refers to 25°C and to the total canopy. We now state that in the table caption.

Avoid overly long sentences, e.g. lines 66–68, which span several embedded clauses. Break these into two sentences for readability.

For this particular sentence (lines 66-68), we don't see a readability issue. Per the reviewer's suggestion, however, we have reviewed the entire manuscript and revised those long sentences to improve readability.